# Single-Pass Contrastive Learning Can Work for Both Homophilic and Heterophilic Graph

**Haonan Wang**                                                    *haonan.wang@u.nus.edu*
*National University of Singapore*

**Jieyu Zhang**                                                    *jieyuz2@cs.washington.edu*
*University of Washington*

**Qi Zhu**                                                         *qiz3@illinois.edu*
*University of Illinois Urbana-Champaign*

**Wei Huang**[*]                                                   *wei.huang.vr@riken.jp*
*RIKEN Center for Advanced Intelligence Project*

**Kenji Kawaguchi**                                                *kenji@nus.edu.sg*
*National University of Singapore*

**Xiaokui Xiao**                                                   *xkxiao@nus.edu.sg*
*National University of Singapore*

**Reviewed on OpenReview:** *https://openreview.net/forum?id=244KePnO9i*

## Abstract

Existing graph contrastive learning (GCL) techniques typically require two forward passes for a single instance to construct the contrastive loss, which is effective for capturing the low-frequency signals of node features. Such a dual-pass design has shown empirical success on homophilic graphs, but its effectiveness on heterophilic graphs, where directly connected nodes typically have different labels, is unknown. In addition, existing GCL approaches fail to provide strong performance guarantees. Coupled with the unpredictability of GCL approaches on heterophilic graphs, their applicability in real-world contexts is limited. Then, a natural question arises: *Can we design a GCL method that works for both homophilic and heterophilic graphs with a performance guarantee?* To answer this question, we theoretically study the concentration property of features obtained by neighborhood aggregation on homophilic and heterophilic graphs, introduce the *single-pass augmentation-free* graph contrastive learning loss based on the property, and provide performance guarantees for the minimizer of the loss on downstream tasks. As a direct consequence of our analysis, we implement the **S**ingle-**P**ass **G**raph **C**ontrastive **L**earning method (SP-GCL). Empirically, on 14 benchmark datasets with varying degrees of homophily, the features learned by the SP-GCL can match or outperform existing strong baselines with significantly less computational overhead, which demonstrates the usefulness of our findings in real-world cases. The code is available at https://github.com/haonan3/SPGCL.

## 1 Introduction

Graph Contrastive Learning (GCL) (Xie et al., 2021), as an approach for learning better representation without the demand of manual annotations, has attracted great attention in recent years. Existing works of GCL could be roughly divided into two categories according to whether or not a graph augmentation is

---

[*]Corresponding Author

employed. First, the *augmentation-based* GCL (You et al., 2020; Peng et al., 2020; Hassani & Khasahmadi, 2020; Zhu et al., 2021a;b; 2020d;c; Thakoor et al., 2021) follows the initial exploration of contrastive learning in the visual domain (Chen et al., 2020; He et al., 2020) and involves pre-specified graph augmentations (Zhu et al., 2021a); specifically, these methods encourage representations of the same node encoded from *two augmentation* views to contain as less information about the way the inputs are transformed as possible during training, *i.e.*, to be invariant to a set of manually specified transformations. Second, *augmentation-free* GCL (Lee et al., 2021; Xia et al., 2022) follows the recent bootstrapped framework (Grill et al., 2020) and constructs different views through *two encoders* of different updating strategies and pushes together the representations of the same node/class. For both categories, there has been notable success on homophilic graphs (where directly linked nodes tend to have similar features or same class labels).

However, real-world graphs do not necessarily adhere to the homophily assumption, but may instead exhibit heterophily, in which directly connected nodes have contrasting characteristics and different class labels. In addition, earlier research shows the importance of high-frequency information for heterophilic graphs (Bo et al., 2021), whereas the dual-pass GCL approaches tend to capture low-frequency information, which has been demonstrated in both theoretical analysis and empirical observations (Liu et al., 2022; Wang et al., 2022a). Moreover, the homophily degree is specified by the node labels, which are difficult to obtain in real-world situations; this makes it infeasible to determine if it is appropriate to apply a homophilic-customized GCL algorithm to a given graph. Coupled with the absence of a strong performance guarantee, the applicability of GCL methods in a variety of real-world scenarios is restricted. Therefore, in this paper, we ask the following question:

> *Can one design a simple graph contrastive learning method that is effective on both homophilic and heterophilic graphs with performance guarantee?*

We provide an affirmative answer to this question both theoretically and empirically. First, we study the neighborhood aggregation mechanism on a homophilic/heterophilic graph and present the concentration property of the aggregated features that is independent of the graph type. By exploiting such a property, we introduce the single-pass augmentation-free graph contrastive loss, and show that its minimizer is equivalent to that of Matrix Factorization (MF) over the transformed graph where the edges are constructed based on the aggregated features. In turn, the transformed graph introduced conceptually is able to help us illustrate and derive the theoretical guarantee for the performance of the learned representations in the down-streaming node classification task.

To verify our theoretical findings, we present a direct implementation of our analysis, **S**ingle-**P**ass **G**raph **C**ontrastive **L**earning (SP-GCL). Experimental results show that SP-GCL achieves competitive performance on 8 homophilic graph benchmarks and outperforms state-of-the-art GCL algorithms on 6 heterophilic graph benchmarks with a substantial margin. Besides, we analyze the computational complexity of SP-GCL and empirically demonstrate a significant reduction of computational overhead brought by SP-GCL. Coupling with extensive ablation studies, we demonstrate that the techniques derived from our theoretical analysis are effective for real-world cases. Our contribution could be summarized as:

- We demonstrate that the concentration attribute of representations derived by neighborhood feature aggregation is independent of graph type, which in turn drives our novel single-pass graph contrastive learning loss. A direct consequence is a *single-pass augmentation-free* graph contrastive learning method, SP-GCL, without relying on graph augmentations.

- We derive a theoretical guarantee for the node embedding obtained by optimizing the graph contrastive learning loss in the downstream node classification task.

- Experimental results show that without complex designs, compared with SOTA GCL methods, SP-GCL achieves competitive or better performance on 8 homophilic graph benchmarks and 6 heterophilic graph benchmarks, with significantly less computational overhead.

## 2 Related Work

**Graph contrastive learning.** Existing graph contrastive learning methods can be categorized into augmentation-based and augmentation-free methods, according to whether or not the graph augmentation techniques are employed during training. The augmentation-based methods (You et al., 2020; Peng et al., 2020; Hassani & Khasahmadi, 2020; Zhu et al., 2021a;b; 2020d; Thakoor et al., 2021; Zhu et al., 2020c) encourage the target graph encoder to be invariant to the manually specified graph transformations. Therefore, the design of graph augmentation is critical to the success of augmentation-based GCL. We summarized the augmentation methods commonly used by recent works in Table 11 of Appendix D. Other works (Lee et al., 2021; Xia et al., 2022) try to get rid of the manual design of augmentation strategies, following the bootstrapped framework (Grill et al., 2020). They construct different views through two graph encoders updated with different strategies and push together the representations of the same node/class from different views. In both categories, those existing GCL methods require two graph forward-pass. Specifically, two augmented views of the same graph will be encoded separately by the same or two graph encoders for augmentation-based GCLs and the same graph will be encoded with two different graph encoders for augmentation-free GCLs. Besides, the theoretical analysis for the performance of GCL in the downstream tasks is still lacking. Although several efforts have been made in the visual domain (Arora et al., 2019; Lee et al., 2020; Tosh et al., 2021; HaoChen et al., 2021), the analysis for image classification cannot be trivially extended to graph setting, since the non-Euclidean graph structure is far more complex.

**Heterophilic graph.** Recently, the heterophily has been recognized as an important issue for graph neural networks, which is outlined by Pei et al. (2020) firstly. To make graph neural networks able to generalize well on the heterophilic graph, several efforts have been done from both the spatial and spectral perspectives (Pei et al., 2020; Abu-El-Haija et al., 2019; Zhu et al., 2020a; Chien et al., 2020; Li et al., 2021; Bo et al., 2021). Firstly, Chien et al. (2020) and Bo et al. (2021) analyze the necessary frequency component for GNNs to achieve good performance on heterophilic graphs and propose methods that are able to utilize high-frequency information. From the spatial perspective, several graph neural networks are designed to capture important dependencies between distant nodes (Pei et al., 2020; Abu-El-Haija et al., 2019; Bo et al., 2021; Zhu et al., 2020a). Although those methods have shown their effectiveness on heterophilic graphs, human annotations are required to guide the learning of neural networks. In addition, from the network embedding view, recent work (Zhong et al., 2022) proposed a method to learn the node embedding for heterophilic graph with self-supervised learning objective.

## 3 Preliminary

**Notation.** Let $\mathcal{G} = (\mathcal{V}, \mathcal{E})$ denote an undirected graph, where $\mathcal{V} = \{v_i\}_{i \in [N]}$ and $\mathcal{E} \subseteq \mathcal{V} \times \mathcal{V}$ denote the node set and the edge set respectively. We denote the number of nodes and edges as $N$ and $E$, and the label of nodes as $\mathbf{y} \in \mathbb{R}^N$, in which $y_i \in \{1, \ldots, c\}$ and $c \geq 2$ is the number of classes. The associated node feature matrix denotes as $\mathbf{X} \in \mathbb{R}^{N \times F}$, where $\mathbf{x}_i \in \mathbb{R}^F$ is the feature of node $v_i \in \mathcal{V}$ and $F$ is the input feature dimension. We denote the adjacent matrix with self-loop as $\mathbf{A} \in \{0, 1\}^{N \times N}$, where $\mathbf{A}_{ii} = 1$ and $\mathbf{A}_{ij} = 1$ if $(v_i, v_j) \in \mathcal{E}$; and the corresponding degree matrix as $\mathbf{D} = \text{diag}(d_1, \ldots, d_N)$, $d_i = \sum_j \mathbf{A}_{i,j}$. Our objective is to unsupervisedly learn a GNN encoder $f_\theta : \mathbf{X}, \mathbf{A} \to \mathbb{R}^{N \times K}$ receiving the node features and graph structure as input, that produces node representations in low dimensionality, i.e., $K \ll F$. The representations can benefit the downstream supervised or semi-supervised tasks, e.g., node classification.

**Homophilic and heterophilic graph.** Various metrics have been proposed to measure the homophily degree of a graph (Zhu et al., 2020b; Pei et al., 2020). In this work, we adopt two representative metrics, namely, node homophily and edge homophily. The edge homophily (Zhu et al., 2020b) is the proportion of edges that connect two nodes of the same class: $h_{edge} = \frac{|\{(v_i, v_j) : (v_i, v_j) \in \mathcal{E} \wedge y_i = y_j\}|}{E}$. And the node homophily (Pei et al., 2020) is defined as, $h_{node} = \frac{1}{N} \sum_{v_i \in \mathcal{V}} \frac{|\{v_j : (v_i, v_j) \in \mathcal{E} \wedge y_i = y_j\}|}{|\{v_j : (v_i, v_j) \in \mathcal{E}\}|}$, which evaluates the average proportion of edge-label consistency of all nodes. They are all in the range of $[0, 1]$ and a value close to 1 corresponds to strong homophily while a value close to 0 indicates strong heterophily. As conventional, we refer the graph with high homophily degree as homophilic graph, and the graph with a low homophily degree as heterophilic graph. And we provided the homophily degree of graph considered in this work in Table 8 of Section A.1.

# 4 Theoretical Analyses

In this section, we first show the property of node representations obtained through neighbor aggregation (Theorem 1). Then, based on the property, we introduce the single-pass graph contrastive loss (Equation (4)), where the contrastive pairs are made using the node representations instead of the augmented graph. With Lemma 1, we bridge the graph contrastive loss and the Matrix Factorization. Then, leveraging the analysis for matrix factorization, we obtain the performance guarantee for the embedding learned by SP-GCL in the downstream node classification task (Theorem 3).

## 4.1 Analysis of Aggregated Features

To obtain analytic and conceptual insights into the aggregated features, we first describe the graph data we considered. Consider the GNN model:

$$\mathbf{Z} = \sigma(\mathbf{D}^{-1}\mathbf{A}\mathbf{X}\ \mathbf{W}) \in \mathbb{R}^{N \times K}$$

where $\sigma$ represents the nonlinear activation function, and $\mathbf{W} \in \mathbb{R}^{F \times K}$ is the weight, $\mathbf{x}_i \in \mathbb{R}^F$ represents the node features of $i$-th input node (i.e., the transpose of the $i$-th row of $\mathbf{X}$). Define $\mathbf{w}_k \in \mathbb{R}^F$ to be the $k$-th column of $\mathbf{W}$. The assumption on the distribution of graph is given as follows:

**Assumption 1** *Consider a node $v_i$ with label $y_i$. Let its node feature $\mathbf{x}_i$ be sampled from the conditional distribution $P_{\mathbf{x}|y}$. Additionally, assume that the labels of $v_i$'s neighbors are independently sampled according to the neighbor distribution $P_{y_i}$. We further assume the existence of $R \in \mathbb{R}^+$, such that $\forall v_i \in \mathcal{V}, \|\mathbf{x}_i\|_2 \leq R$.*

**Remark.** The above assumption means that the label of a neighbor is only determined by the label of the central node. This is feasible for both homophily and heterophily. Specifically, within the framework of our stated assumption, if the neighbor distribution $P_{y_i}$ exhibits a bias towards a high probability of sampling a label identical to $y_i$ (the label of the central node), then a homophily trend emerges. In such cases, nodes exhibit a higher likelihood of forming connections with other nodes that share the same label, thereby leading to a graph characterized by homophily. Conversely, if the probability distribution $P_{y_i}$ tends to sample labels that are different from $y_i$, the graph displays a tendency towards heterophily. This is because nodes are more likely to form connections with nodes that have different labels, echoing the definition of a heterophilic graph. A representative example of a graph generative model adhering to Assumption 1 is the contextual stochastic block model (Deshpande et al., 2018; Lu & Sen, 2020). Specifically, in this model, the adjacency matrix $\mathbf{A}$ follows a Bernoulli distribution. We have $\mathbf{A}_{ij} = a_{ij}$, $a_{ij} \sim \text{Ber}(p)$ when $y_i = y_j$, and $a_{ij} \sim \text{Ber}(s)$ when $y_i = -y_j$. Additionally, the node feature is defined as $\mathbf{x} = y\boldsymbol{\mu} + \boldsymbol{\xi}$, where $\boldsymbol{\mu} \in \mathbb{R}^F$ and $\boldsymbol{\xi} \sim \mathcal{N}(\mathbf{0}, \mathbf{I}) \in \mathbb{R}^F$.

Assumption 1 has been extensively utilized in various studies, especially those aiming to theoretically comprehend graph neural networks, as exemplified by Huang et al. (2023); Ma et al. (2021); Baranwal et al. (2021; 2023). This demonstrates the comprehensive applicability and relevance of this assumption across diverse applications. Furthermore, we provide our assumption on the activation function:

**Assumption 2** *Assume that the nonlinear activation function $\sigma$ is homogeneous of degree $\tau$ for some $\tau \geq 1$ and $\sigma(q) \leq cq$ for some $c \geq 0$.*

For example, ReLU activation satisfies Assumption 2 with $\tau = c = 1$. Also, it covers the result of the linear model too since if $\sigma$ is identity (i.e., linear model), then $\tau = c = 1$. With above assumptions, we present the following Theorem 1, where we denote the nodes representation obtained through one layer of graph neural network as $\mathbf{Z}$, and $\mathbf{Z}_i$ represents the learned embedding with respect to node $v_i$.

**Theorem 1 (Concentration Property of Aggregated Features)** *Under Assumptions 1 and 2, for any $\boldsymbol{W} \in \mathbb{R}^{F \times K}$, the following statements holds:*

*1. For any pair $(i, i') \in \mathcal{E}$, such that $y_i = y_{i'}$,*

$$\mathbb{E}[\mathbf{Z}_i] - \mathbb{E}[\mathbf{Z}_{i'}] = 0. \tag{1}$$

2. *With probability at least $1 - \delta$,*

$$\|\mathbb{E}[\mathbf{Z}_i] - \mathbf{Z}_i\|_\infty \leq cR \left( \max_{k \in [K]} \|\boldsymbol{w}_k\|_2 \right) \sqrt{\frac{2\ln(2K/\delta)}{D_{ii}^{2\tau - 1}}}. \tag{2}$$

3. *For any node pair $(i, j)$, with probability at least $1 - \delta'$,*

$$|\mathbb{E}[\mathbf{Z}_i^\top \mathbf{Z}_j] - \mathbf{Z}_i^\top \mathbf{Z}_j| \leq c^2 R^2 \sum_{k=1}^{K} \|\boldsymbol{w}_k\|_2^2 \sqrt{\frac{2\ln(2/\delta')}{D_{ii}^{2\tau - 1} D_{jj}^{2\tau - 1}}}. \tag{3}$$

**Remark.** The above theorem indicates that, for any graph following the graph data assumption: ***(i)*** in expectation, nodes with the same label have the same representation, Equation (1); ***(ii)*** the representations of nodes with the same label tend to concentrate onto a certain area in the embedding space, which can be regarded as the inductive bias of the neighborhood aggregation mechanism, Equation (2); ***(iii)*** the inner product of hidden representations approximates to its expectation with a high probability, Equation (3). (Proof is in Appendix C.1.)

Unlike the preliminary concentration study presented in Ma et al. (2021), our work advances the theoretical understanding of aggregated features in the non-linear setting, making it more applicable to the types of Graph Neural Network (GNN) models commonly used in real-world scenarios. We substantiate our theoretical contributions with empirical evidence, specifically validating the derived concentration property (as stated in Theorem 1) using multi-class, real-world graph data in conjunction with a non-linear multi-layer graph neural network. The outcomes, as discussed in Section 6.4 and summarized in Table 5, are in accord with our analytical framework. This result is also consistent with findings from other multi-layer graph convolution studies, such as the one by Baranwal et al. (2022). Furthermore, we extend the discourse by offering a comprehensive discussion of how the concentration property has been observed empirically in recent works, including those by Wang et al. (2022b) and Trivedi et al. (2022), in Section 7.

### 4.2 Single-Pass Graph Contrastive Loss

In order to learn a more compact and linearly separable embedding space, we introduce the single-pass graph contrastive loss based on the property of aggregated features. Taking advantage of the concentration attribute explicitly, we consider nodes with a small distance in the embedding space to be positive pairs, and we characterize randomly chosen nodes as negative pairs by convention (Arora et al., 2019; Chen et al., 2020). Also, a theoretical discussion is provided for this intuitive design (Theorem 2). We formally define the positive and negative pairs to introduce the loss. For node $v_i \in \mathcal{V}$, we draw the positive node $v_{i^+}$ uniformly from the set $S_{pos}^i$, $v_{i^+} \sim \text{Uni}(S_{pos}^i)$. The set $S_{pos}^i$ is consisted by the $K_{pos}$ nodes closest to node $v_i$. Concretely, $S_{pos}^i = \{v_i^1, v_i^2, \ldots, v_i^{K_{pos}}\} = \arg\max_{v_j \in \mathcal{V}} (\mathbf{Z}_i^\top \mathbf{Z}_j / \|\mathbf{Z}_i\|_2 \|\mathbf{Z}_j\|_2, K_{pos})$, where $K_{pos} \in \mathbb{Z}^+$ and $\arg\max(\cdot, K_{pos})$ denotes the operator for the top-$K_{pos}$ selection. Two nodes, $(v_i, v_{i^+})$, form a positive pair. For nodes $v_j$ and $v_k$, sampled independently from the node set $\mathcal{V}$, can be regarded as a negative pair $(v_j, v_k)$. Following the insight of contrastive learning (Wang & Isola, 2020), similar sample pairs stay close to each other while dissimilar ones are far apart, then the Single-Pass Graph Contrastive Loss is defined as,

$$\mathcal{L}_{\text{SP-GCL}} = -2 \mathop{\mathbb{E}}_{\substack{v_i \sim \text{Uni}(\mathcal{V}) \\ v_{i^+} \sim \text{Uni}(S_{pos}^i)}} \left[ \mathbf{Z}_i^\top \mathbf{Z}_{i^+} \right] + \mathop{\mathbb{E}}_{\substack{v_j \sim \text{Uni}(\mathcal{V}) \\ v_k \sim \text{Uni}(\mathcal{V})}} \left[ \left( \mathbf{Z}_j^\top \mathbf{Z}_k \right)^2 \right]. \tag{4}$$

Compared with conventional Graph Contrastive Learning (GCL) methods (You et al., 2020; Peng et al., 2020; Hassani & Khasahmadi, 2020; Zhu et al., 2021a;b; 2020d;c; Thakoor et al., 2021), which heavily relied on graph augmentations for defining their contrastive losses (The various augmentation techniques employed by these methods are exhaustively detailed in the Table 11 of Appendix), our proposed approach circumvents the need for explicit augmentation techniques, a deviation that simplifies the contrastive learning process and potentially reduces the computational overhead. In addition, our approach stands distinct from other augmentation-free methods such as those proposed by Lee et al. (2021); Xia et al. (2022). These methods construct negative pairs through a dual forward pass, leveraging a slower-moving average to create different

views. While this strategy certainly has its merits, it can introduce additional complexity, e.g. two forward pass, to the learning process.

The proposed SP-GCL is single-pass and augmentation-free. In contrast, it manages to avoid both the need for explicit augmentation techniques and the dual-forward-pass strategy. This unique approach does not rely on multiple views or passes to construct negative pairs, thereby streamlining the contrastive learning process. Our findings suggest that this approach is not only feasible but also effective across both homophilic and heterophilic graphs.

### 4.3 Performance Guarantee for Learning Linear Classifier

For the convenience of analysis, we first introduce the concept of *transformed graph* as follows, which is constructed based on the original graph and the selected positive pairs. *Note, the transformed graph is a concept only used to illustrate the theoretical derivation, instead of the actual computation.*

**Definition 1 (Transformed Graph)** *Given the original graph $\mathcal{G}$ and its node set $\mathcal{V}$, the transformed graph, $\widehat{\mathcal{G}}$, has the same node set $\mathcal{V}$ but with the selected positive pairs as the edge set, $\widehat{\mathcal{E}} = \bigcup_i \{(v_i, v_i^k)|_{k=1}^{K_{pos}}\}$.*

**Remark.** The transformed graph is formed by positive pairs selected based on aggregated features. Coupling with the *Concentration Property of Aggregated Features* (Theorem 1), the transformed graph tends to have a larger homophily degree than the original graph. We provide more empirical verifications in Section 6.4.

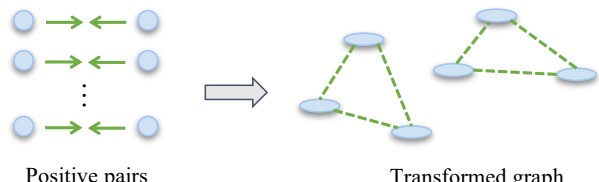

Positive pairs                  Transformed graph

Figure 1: Transformed Graph formed with Positive Pairs.

The transformed graph is illustrated in Figure 1. We denote the adjacency matrix of transformed graph as $\widehat{\mathbf{A}} \in \{0,1\}^{N \times N}$, the number of edges as $\hat{E}$, and the symmetric normalized matrix of transformed graph as $\widehat{\mathbf{A}}_{sym} = \widehat{\mathbf{D}}^{-1/2}\widehat{\mathbf{A}}\widehat{\mathbf{D}}^{-1/2}$, where $\widehat{\mathbf{D}} = \text{diag}\left(\hat{d}_1, \ldots, \hat{d}_N\right)$, $\hat{d}_i = \sum_j \widehat{\mathbf{A}}_{i,j}$. Correspondingly, we denote the symmetric normalized Laplacian as $\widehat{\mathbf{L}}_{sym} = \mathbf{I} - \widehat{\mathbf{A}}_{sym} = \mathbf{U}\mathbf{\Lambda}\mathbf{U}^\top$. Here $\mathbf{U} \in \mathbb{R}^{N \times N} = [\mathbf{u}_1, \ldots, \mathbf{u}_N]$, where $\mathbf{u}_i \in \mathbb{R}^N$ denotes the $i$-th eigenvector of $\widehat{\mathbf{L}}_{sym}$ and $\mathbf{\Lambda} = \text{diag}\left(\lambda_1, \ldots, \lambda_N\right)$ is the corresponding eigenvalue matrix. $\lambda_1$ and $\lambda_N$ be the smallest and largest eigenvalue respectively.

Then we show that optimizing a model with the contrastive loss (Equation (4)) is equivalent to the matrix factorization over the transformed graph, as stated in the following Lemma:

**Lemma 1** *Denote the learnable embedding for matrix factorization as $\mathbf{F} \in \mathbb{R}^{N \times K}$. Let $\mathbf{F}_i = F_\psi(v_i)$. Then, the matrix factorization loss function $\mathcal{L}_{\text{mf}}$ is equivalent to the contrastive loss, Equation (4), up to an additive constant:*

$$\mathcal{L}_{\text{mf}}(\mathbf{F}) = \left\|\widehat{\mathbf{A}}_{sym} - \mathbf{F}\mathbf{F}^\top\right\|_F^2 = \mathcal{L}_{\textit{SP-GCL}} + const \tag{5}$$

The above lemma bridges the graph contrastive learning and the graph matrix factorization and therefore allows us to provide the performance guarantee of SP-GCL by leveraging the power of matrix factorization. We leave the derivation in Appendix C.2. With the Theorem 1 and Lemma 1, we arrive at a conclusion about the expected value of the inner product of embeddings (More details in Appendix C.3):

**Theorem 2** *For a graph $G$ following the graph data assumption (Assumptions 1), then when the optimal of the contrastive loss is achieved, i.e., $\mathcal{L}_{\text{mf}}(\mathbf{F}^*) = 0$, we have,*

$$\mathbb{E}[\mathbf{Z}_i^{*\top}\mathbf{Z}_j^*|_{y_i=y_j}] - \mathbb{E}[\mathbf{Z}_i^{*\top}\mathbf{Z}_j^*|_{y_i \neq y_j}] \geq 1 - \bar{\phi}, \tag{6}$$

*where $\bar{\phi} = \mathbb{E}_{v_i, v_j \sim Uni(\mathcal{V})}\left(\widehat{\mathbf{A}}_{i,j} \cdot \mathbb{1}[y_i \neq y_j]\right)$.*

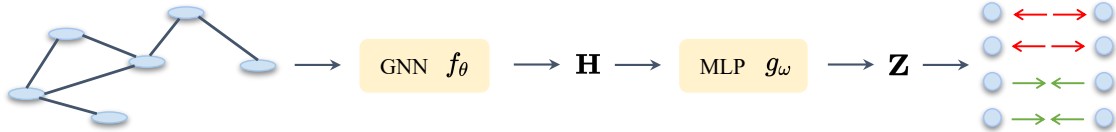

Figure 2: Overview of SP-GCL. The graph data is encoded by a graph neural network $f_\theta$ and a following projection head $g_\omega$. The contrastive pairs are constructed based on the representation $\mathbf{Z}$.

**Remark.** The $1 - \bar{\phi}$ means the probability of an edge connecting two nodes from the same class, which can be regarded as the edge homophily of the transformed graph. Its value is always greater than 0. The theorem implies that a pair of nodes coming from the same class are likely have larger inner product similarity than a pair of nodes coming from different classes.

Finally, with the Theorem 1 and Theorem 2, we obtain a performance guarantee for node embeddings learned by SP-GCL with a linear classifier in the downstream task (More details in Appendix C.4):

**Theorem 3** *Let $\mathbf{Z}^*$ be a minimizer of the contrastive loss, $\mathcal{L}_{SP\text{-}GCL}$. Then there exists a linear classifier $\mathbf{B}^* \in \mathbb{R}^{c \times K}$ with norm $\|\mathbf{B}^*\|_F \leq 1/(1 - \lambda_K)$ such that, with a probability at least $1 - \delta'$,*

$$\mathbb{E}_{v_i} \left[ \left\| \vec{y}_i - \mathbf{B}^{*\top} \mathbf{Z}^* \right\|_2^2 \right] \leq \frac{\bar{\phi}}{\hat{\lambda}_{K+1}} + c^2 R^2 C_{degree} \sqrt{\frac{2 \ln(2/\delta')}{\hat{\lambda}_{K+1}^2}} \sum_{k=1}^{K} \|\mathbf{w}_k\|_2^2, \tag{7}$$

*where $\vec{y}_i$ is the one-hot embedding of the label of node $v_i$, $C_{degree} = \mathbb{E}_{v_i \sim Uni(\mathcal{V})} \left[ (D_{ii})^{-(2\tau-1)} \right] > 0$ is determined by node degree of the input graph and invariant to the learning process. The positive real constant $R$ and $c$ are defend in Assumption 1 and 2 respectively. $\mathbf{Z}^*$ is obtained through the non-linear graph neural network, and the nonlinearity follows the assumption 2. $\hat{\lambda}_i$ are the $i$ smallest eigenvalues of the symmetrically normalized Laplacian matrix of the transformed graph.*

**Remark.** Our theorem elucidates a noteworthy correlation between a graph's average degree and its associated classification error. Specifically, the theorem posits that graphs with higher average degrees yield a reduced $C_{\text{degree}}$, consequently enhancing node classification accuracy. This assertion is substantiated by empirical data presented in Tables 1 and 2. Although the relationship between the classification accuracy of SP-GCL and the average degree of the graphs (as detailed in Table 8) does not follow a strictly linear pattern, a discernible trend is evident. In particular, datasets with elevated average degrees—such as Amazon Computer, Photo, Coauthor CS, and the Physics dataset—consistently demonstrate superior classification performance. Moreover, the theorem suggests that a smaller value of $\sum_{k=1}^{K} \|\mathbf{w}_k\|_2^2$ leads to better classification accuracy. This prediction prompted the inclusion of batch normalization in the SP-GCL implementation, providing more effective control over the magnitude of the weight vectors and hence improving our node classification procedure. The study of batch normalization's effect in Section 6.3 aligns with the theorem's prediction.

## 5 Single-Pass Augmentation-Free Graph Contrastive Learning (SP-GCL)

As a direct consequence of our theory, we introduce the Single-Pass Augmentation-Free Graph Contrastive Learning (SP-GCL). Instead of relying on the graph augmentation function or the exponential moving average, our novel learning framework only forwards a single time and constructs contrastive pairs based on the aggregated features. As we shall see, this extremely simple, theoretically grounded strategy also achieves superior performance in practice than dual-pass graph contrastive learning methods on homophilic and heterophilic graph benchmarks.

According to the results of the analysis, for each class, the embedding generated through neighbor aggregation will converge on the expected class embedding. We build the self-supervision signal based on the discovered embedding and offer a novel single-pass graph contrastive learning framework without augmentation, SP-GCL, that picks similar nodes as positive node pairings. As shown in Figure 2, in each iteration, the proposed framework firstly encodes the graph with a graph encoder $f_\theta$ denoted by $\mathbf{H} = f_\theta(\mathbf{X}, \mathbf{A})$. Then,

a projection head with $L2$-normalization, $g_\omega$, is employed to project the node embedding into the hidden representation $\mathbf{Z} = g_\omega(\mathbf{H})$. To scale up SP-GCL on large graphs, the node pool, $P$, are constructed by the $T$-hop neighborhood of $b$ nodes (the seed node set $S$) uniformly sampled from $\mathcal{V}$. For each seed node $v_i \in S$, the top-$K_{pos}$ nodes with highest similarity from the node pool are selected as positive set for it which denote as $\tilde{S}^i_{pos} = \arg\max_{v_j \in P} \left( \mathbf{Z}_i^\top \mathbf{Z}_j / \|\mathbf{Z}_i\|_2 \|\mathbf{Z}_j\|_2, K_{pos} \right)$, and $S^i_{neg} = \{v_i^1, v_i^2, \ldots, v_i^{K_{neg}}\} \sim \mathcal{V}$, $S^i_{neg} \subseteq \mathcal{V}$. Concretely, the framework is optimized with the following objective:

$$\widehat{\mathcal{L}}_{\text{SP-GCL}} = -\frac{2}{N \cdot K_{pos}} \sum_{v_i \in \mathcal{V}} \sum_{v_{i+} \in \tilde{S}^i_{pos}} \left[ \mathbf{Z}_i^\top \mathbf{Z}_{i+} \right] + \frac{1}{N \cdot K_{neg}} \sum_{v_j \in \mathcal{V}} \sum_{v_k \in S^i_{neg}} \left[ \left( \mathbf{Z}_j^\top \mathbf{Z}_k \right)^2 \right]. \tag{8}$$

With $T = 1$ or $|P| = N$, the empirical contrastive loss is an unbiased estimation of the Equation (4). With $T > 1$ and $|P| < N$, the sampling will bias towards nodes with large degree. Overall, the training algorithm SP-GCL is summarized in Algorithm 1.

---

**Algorithm 1** Single-Pass Augmentation-Free Graph Contrastive Learning (SP-GCL).

---

**Input:** Graph neural network $f_\theta$, MLP projection head $g_\omega$, input adjacency matrix $\mathbf{A}$, node features $\mathbf{X}$, batch size $b$, number of hops $T$, number of positive nodes $K_{pos}$.
**for** epoch $\leftarrow 1, 2, \cdots$ **do**
    1. Obtain the node embedding, $\mathbf{H} = f_\theta(\mathbf{X}, \mathbf{A})$.
    2. Obtain the hidden representation, $\mathbf{Z} = g_\omega(\mathbf{H})$.
    3. Sample $b$ nodes as seed node set $S$ and construct the node pool $P$ with the $T$-hop neighbors of each node in the set $S$.
    4. Select top-$K_{pos}$ similar nodes for every $v_i \in S$ to form the positive node set $\tilde{S}^i_{pos}$.
    5. Compute the contrastive objective with Eq. (8) and update parameters by applying stochastic gradient.
**end for**
**return** Final model $f_\theta$.

---

Although our theoretical analysis indicates that SP-GCL can ensure performance for both homophilic and heterophilic graphs, it is yet unknown whether the proposed strategy is effective in real-world situations. In the subsequent part, we empirically validate the effectiveness of the method and the applicability of our conclusions across a wide variety of graph datasets.

# 6 Experiments

We conduct experiments to evaluate the SP-GCL by answering the following questions:

- **RQ1**: How does the SP-GCL perform compared with other GCL methods on homophilic and heterophilic graphs?

- **RQ2**: What's the empirical memory consumption of SP-GCL compared with prior works?

- **RQ3**: Whether the learning procedure is stable? And whether node representation have expected behavior during training?

- **RQ4**: How do the specific hyper-parameters of SP-GCL, such as the selection of top-$K$ and sampling hop-$T$, affect its performance?

To answer these questions, we apply supervised approaches, augmentation-based and augmentation-free GCL methods (a total of 14 baseline methods) in accordance with the linear evaluation scheme described in Velickovic et al. (2019). In addition, we utilize 14 benchmarking graph data to assess SP-GCL on diverse datasets with varying scales and features. We present the selected datasets, baselines, evaluation protocol and implementation details in the Appendix A.1, A.2, A.3, and A.4 respectively. In addition, Appendix A contains most details about infrastructures and hyperparameter configurations.

## 6.1 Performance on Homophilic and Heterophilic graph

The homophilic graph benchmarks have been studied by several previous works (Velickovic et al., 2019; Peng et al., 2020; Hassani & Khasahmadi, 2020; Zhu et al., 2020d; Thakoor et al., 2021; Lee et al., 2021). We re-use their configuration and compare SP-GCL with those methods and leave the detailed description about the experiment setting in Appendix A. The result is summarized in Table 1, in which the best performance achieved by self-supervised methods is marked in boldface. Compared with augmentation-based and augmentation-free GCL methods, SP-GCL outperforms previous methods on 2 datasets and achieves competitive performance on the others, which shows the effectiveness of the single-pass contrastive loss on homophilic graphs. We further assess the model performance on heterophilic graph benchmarks that employed in Pei et al. (2020) and Lim et al. (2021a). As shown in Table 2, SP-GCL achieves the best performance on 6 of 6 heterophilic graphs by an evident margin. The above result indicates that, instead of relying on the augmentations which are sensitive to the graph type, **SP-GCL *is able to work well over a wide range of real-world graphs with different homophily degree*.** Additionally, we assess the combination of SP-GCL with three other popular GNNs, namely GAT (Veličković et al., 2017), GIN (Xu et al., 2018), and GraphSAGE (Hamilton et al., 2017). The findings, presented in Table 10 of Appendix B.1, affirm the feasibility of our proposed training approach across diverse graph neural network architectures. For an in-depth discussion on this matter, we direct the readers to Appendix B.1.

Table 1: Graph Contrastive Learning on Homophilic Graphs. The highest performance of unsupervised models is highlighted in boldface. OOM indicates Out-Of-Memory on a 32GB GPU.

| MODEL | CORA | CITESEER | PUBMED | WIKICS | AMZ-COMP. | AMZ-PHOTO | COAUTHOR-CS | COAUTHOR-PHY. | AVG. |
|---|---|---|---|---|---|---|---|---|---|
| MLP | $47.92 \pm 0.41$ | $49.31 \pm 0.26$ | $69.14 \pm 0.34$ | $71.98 \pm 0.42$ | $73.81 \pm 0.21$ | $78.53 \pm 0.32$ | $90.37 \pm 0.19$ | $93.58 \pm 0.41$ | 71.83 |
| GCN | $81.54 \pm 0.68$ | $70.73 \pm 0.65$ | $79.16 \pm 0.25$ | $78.02 \pm 0.11$ | $86.51 \pm 0.54$ | $92.42 \pm 0.22$ | $93.03 \pm 0.31$ | $95.65 \pm 0.16$ | 84.63 |
| DEEPWALK | $70.72 \pm 0.63$ | $51.39 \pm 0.41$ | $73.27 \pm 0.86$ | $74.42 \pm 0.13$ | $85.68 \pm 0.07$ | $89.40 \pm 0.11$ | $84.61 \pm 0.22$ | $91.77 \pm 0.15$ | 77.65 |
| NODE2CEC | $71.08 \pm 0.91$ | $47.34 \pm 0.84$ | $66.23 \pm 0.95$ | $71.76 \pm 0.14$ | $84.41 \pm 0.14$ | $89.68 \pm 0.19$ | $85.16 \pm 0.04$ | $91.23 \pm 0.07$ | 75.86 |
| SELENE | $79.21 \pm 0.76$ | $68.94 \pm 1.22$ | OOM | $77.76 \pm 0.81$ | OOM | $90.61 \pm 0.45$ | OOM | OOM | – |
| GAE | $71.49 \pm 0.41$ | $65.83 \pm 0.40$ | $72.23 \pm 0.71$ | $73.97 \pm 0.16$ | $85.27 \pm 0.19$ | $91.62 \pm 0.13$ | $90.01 \pm 0.71$ | $94.92 \pm 0.08$ | 80.66 |
| VGAE | $77.31 \pm 1.02$ | $67.41 \pm 0.24$ | $75.85 \pm 0.62$ | $75.56 \pm 0.28$ | $86.40 \pm 0.22$ | $92.16 \pm 0.12$ | $92.13 \pm 0.16$ | $94.46 \pm 0.13$ | 82.66 |
| DGI | $82.34 \pm 0.71$ | $71.83 \pm 0.54$ | $76.78 \pm 0.31$ | $75.37 \pm 0.13$ | $84.01 \pm 0.52$ | $91.62 \pm 0.42$ | $92.16 \pm 0.62$ | $94.52 \pm 0.47$ | 83.58 |
| GMI | $82.39 \pm 0.65$ | $71.72 \pm 0.15$ | $79.34 \pm 1.04$ | $74.87 \pm 0.13$ | $82.18 \pm 0.27$ | $90.68 \pm 0.18$ | OOM | OOM | – |
| MVGRL | $\mathbf{83.45 \pm 0.68}$ | $\mathbf{73.28 \pm 0.48}$ | $80.09 \pm 0.62$ | $77.51 \pm 0.06$ | $87.53 \pm 0.12$ | $91.74 \pm 0.08$ | $92.11 \pm 0.14$ | $95.33 \pm 0.05$ | 85.13 |
| GRACE | $81.92 \pm 0.89$ | $71.21 \pm 0.64$ | $\mathbf{80.54 \pm 0.36}$ | $78.19 \pm 0.10$ | $86.35 \pm 0.44$ | $92.15 \pm 0.25$ | $92.91 \pm 0.20$ | $95.26 \pm 0.22$ | 84.82 |
| GCA | $82.07 \pm 0.10$ | $71.33 \pm 0.37$ | $80.21 \pm 0.39$ | $78.40 \pm 0.13$ | $87.85 \pm 0.31$ | $92.49 \pm 0.11$ | $92.87 \pm 0.14$ | $95.68 \pm 0.05$ | 85.11 |
| BGRL | $81.44 \pm 0.72$ | $71.82 \pm 0.48$ | $80.18 \pm 0.63$ | $76.96 \pm 0.61$ | $89.62 \pm 0.37$ | $93.07 \pm 0.34$ | $92.67 \pm 0.21$ | $95.47 \pm 0.28$ | 85.15 |
| AFGRL | $81.60 \pm 0.54$ | $71.02 \pm 0.37$ | $80.02 \pm 0.48$ | $77.98 \pm 0.49$ | $89.66 \pm 0.40$ | $\mathbf{93.14 \pm 0.36}$ | $\mathbf{93.27 \pm 0.17}$ | $\mathbf{95.69 \pm 0.10}$ | 85.29 |
| SP-GCL | $83.16 \pm 0.13$ | $71.96 \pm 0.42$ | $79.16 \pm 0.73$ | $\mathbf{79.01 \pm 0.51}$ | $\mathbf{89.68 \pm 0.19}$ | $92.49 \pm 0.31$ | $91.92 \pm 0.10$ | $95.12 \pm 0.15$ | $\mathbf{85.31}$ |

Table 2: Graph Contrastive Learning on Heterophilic Graphs. The highest performance of unsupervised models is highlighted in boldface. OOM indicates Out-Of-Memory on a 32GB GPU.

| MODEL | CHAMELEON | SQUIRREL | ACTOR | TWITCH-DE | TWITCH-GAMERS | GENIUS | AVG. |
|---|---|---|---|---|---|---|---|
| MLP | $47.59 \pm 0.73$ | $31.67 \pm 0.61$ | $35.93 \pm 0.61$ | $69.44 \pm 0.67$ | $60.71 \pm 0.18$ | $86.62 \pm 0.11$ | 55.32 |
| GCN | $66.45 \pm 0.48$ | $53.03 \pm 0.57$ | $28.79 \pm 0.23$ | $73.43 \pm 0.71$ | $62.74 \pm 0.03$ | $87.72 \pm 0.18$ | 62.02 |
| DEEPWALK | $43.99 \pm 0.67$ | $30.90 \pm 1.09$ | $25.50 \pm 0.28$ | $70.39 \pm 0.77$ | $61.71 \pm 0.41$ | $68.98 \pm 0.15$ | 50.24 |
| NODE2CEC | $31.49 \pm 1.17$ | $27.64 \pm 1.36$ | $27.04 \pm 0.56$ | $70.70 \pm 1.15$ | $61.12 \pm 0.29$ | $67.96 \pm 0.17$ | 47.65 |
| SELENE | $61.37 \pm 0.67$ | $44.84 \pm 2.01$ | $27.59 \pm 0.59$ | $71.67 \pm 1.87$ | OOM | OOM | – |
| GAE | $39.13 \pm 1.34$ | $34.65 \pm 0.81$ | $25.36 \pm 0.24$ | $67.43 \pm 1.16$ | $56.26 \pm 0.50$ | $83.36 \pm 0.21$ | 51.03 |
| VGAE | $42.65 \pm 1.27$ | $35.11 \pm 0.92$ | $28.43 \pm 0.57$ | $68.62 \pm 1.82$ | $60.70 \pm 0.61$ | $85.17 \pm 0.52$ | 53.44 |
| DGI | $60.27 \pm 0.70$ | $42.22 \pm 0.63$ | $28.30 \pm 0.76$ | $72.77 \pm 1.30$ | $61.47 \pm 0.56$ | $86.96 \pm 0.44$ | 58.66 |
| GMI | $52.81 \pm 0.63$ | $35.25 \pm 1.21$ | $27.28 \pm 0.87$ | $71.21 \pm 1.27$ | OOM | OOM | – |
| MVGRL | $53.81 \pm 1.09$ | $38.75 \pm 1.32$ | $28.64 \pm 0.78$ | $71.86 \pm 1.21$ | OOM | OOM | – |
| GRACE | $61.24 \pm 0.53$ | $41.09 \pm 0.85$ | $28.27 \pm 0.43$ | $72.49 \pm 0.72$ | OOM | OOM | – |
| GCA | $60.94 \pm 0.81$ | $41.53 \pm 1.09$ | $28.89 \pm 0.50$ | $73.21 \pm 0.83$ | OOM | OOM | – |
| BGRL | $64.86 \pm 0.63$ | $46.24 \pm 0.70$ | $28.80 \pm 0.54$ | $73.31 \pm 1.11$ | $60.93 \pm 0.32$ | $86.78 \pm 0.71$ | 60.15 |
| AFGRL | $59.03 \pm 0.78$ | $42.36 \pm 0.40$ | $27.43 \pm 1.31$ | $69.11 \pm 0.72$ | OOM | OOM | – |
| SP-GCL | $\mathbf{65.28 \pm 0.53}$ | $\mathbf{52.10 \pm 0.67}$ | $\mathbf{28.94 \pm 0.69}$ | $\mathbf{73.51 \pm 0.97}$ | $\mathbf{62.04 \pm 0.17}$ | $\mathbf{90.06 \pm 0.18}$ | $\mathbf{61.98}$ |

Table 3: Computational requirements on a set of standard benchmark graphs. OOM indicates running out of memory on a 32GB GPU.

| DATASET | CS. | PHY. | GENIUS | GAMERS. |
|---|---|---|---|---|
| # NODES | 18,333 | 34,493 | 421,961 | 168,114 |
| # EDGES | 327,576 | 991,848 | 984,979 | 6,797,557 |
| GRACE | 13.21 GB | 30.11 GB | OOM | OOM |
| BGRL | 3.10 GB | 5.42 GB | 8.18 GB | 26.22 GB |
| SP-GCL | 2.07 GB | 3.21 GB | 6.24 GB | 22.15 GB |

Table 4: True positive ratio of selected edges at the end of training. The minimal, maximal, average, standard deviation of 20 runs are presented.

| Dataset | $h_{edge}$ of $\widehat{\mathcal{G}}$ |
|---|---|
| Cora | $0.812_\uparrow \pm 0.0022$ (0.810) |
| CiteSeer | $0.691_\downarrow \pm 0.0018$ (0.736) |
| PubMed | $0.819_\uparrow \pm 0.0011$ (0.802) |
| Coauthor CS | $0.883_\uparrow \pm 0.0018$ (0.808) |
| Coauthor Phy. | $0.952_\uparrow \pm 0.0021$ (0.931) |
| Amazon Comp. | $0.866_\uparrow \pm 0.0019$ (0.777) |
| Amazon Photo | $0.908_\uparrow \pm 0.0012$ (0.827) |
| WikiCS | $0.751_\uparrow \pm 0.0027$ (0.654) |
| Chameleon | $0.631_\uparrow \pm 0.0047$ (0.234) |
| Squirrel | $0.526_\uparrow \pm 0.0042$ (0.223) |
| Actor | $0.378_\uparrow \pm 0.0026$ (0.216) |
| Twitch-DE | $0.669_\uparrow \pm 0.0033$ (0.632) |
| Twitch-gamers | $0.617_\uparrow \pm 0.0021$ (0.545) |
| Genius | $0.785_\uparrow \pm 0.0028$ (0.618) |

Table 5: Edge homophily of the transformed graph at the initial stage. Homophily value of original graphs is shown in parentheses.

| Dataset | Min | Max | Avg | Std |
|---|---|---|---|---|
| Cora | 0.817 | 0.836 | 0.826 | 0.0061 |
| CiteSeer | 0.708 | 0.719 | 0.713 | 0.0033 |
| PubMed | 0.820 | 0.832 | 0.825 | 0.0037 |
| Coauthor CS | 0.892 | 0.905 | 0.901 | 0.0025 |
| Coauthor Phy. | 0.952 | 0.963 | 0.959 | 0.0028 |
| Amazon Comp. | 0.874 | 0.896 | 0.878 | 0.0076 |
| Amazon Photo | 0.903 | 0.926 | 0.916 | 0.0054 |
| WikiCS | 0.757 | 0.771 | 0.761 | 0.0040 |
| Chameleon | 0.703 | 0.785 | 0.711 | 0.0183 |
| Squirrel | 0.529 | 0.541 | 0.530 | 0.0024 |
| Actor | 0.388 | 0.399 | 0.396 | 0.0016 |
| Twitch-DE | 0.693 | 0.726 | 0.703 | 0.0073 |
| Twitch-gamers | 0.620 | 0.633 | 0.627 | 0.0022 |
| Genius | 0.786 | 0.797 | 0.790 | 0.0029 |

## 6.2 Computational Complexity Analysis

In order to illustrate the advantages of SP-GCL, we provide a brief comparison of the time and space complexities between SP-GCL, the previous strong contrastive method, GCA (Zhu et al., 2021b), and the memory-efficient contrastive method, BGRL (Thakoor et al., 2021). Consider a graph with $N$ nodes and $E$ edges, and a graph neural network (GNN), $f$, that compute node embeddings in time and space $O(N + E)$. BGRL performs four GNN computations per update step, in which twice for the target and online encoders, and twice for each augmentation, and a node-level projection; GCA performs two GNN computations (once for each augmentation), plus a node-level projection. Both methods backpropagate the learning signal twice (once for each augmentation), and we assume the backward pass to be approximately as costly as a forward pass. Both of them will compute the augmented graphs by feature masking and edge masking on the fly, the cost for augmentation computation is nearly the same. Thus the total time and space complexity per update step for BGRL is $6C_{encoder}(E + N) + 4C_{proj}N + C_{prod}N + C_{aug}$ and $4C_{encoder}(E + N) + 4C_{proj}N + C_{prod}N^2 + C_{aug}$ for GCA. The $C_{prod}$ depends on the dimension of node embedding and we assume the node embeddings of all the methods with the same size. For our proposed method, only one GNN encoder is employed and we compute the inner product of $b$ nodes to construct positive samples and $K_{pos}$ and $K_{neg}$ inner product for the loss computation. Then for SP-GCL, we have: $2C_{encoder}(E + N) + 2C_{proj}N + C_{prod}(K_{pos} + K_{neg})^2$, which is significantly lower than BGRL and GCA. We empirically measure the peak of GPU memory usage of SP-GCL, GCA and BGRL. As a fair comparison, we set the embedding size as 128 for all those methods on the four datasets and keep the other hyper-parameters of the three methods the same as the main experiments. As shown by Table 3, *__the computational overhead of__ SP-GCL *is much less than the previous methods*.

## 6.3 Effective Way to Keep the Summation of the $l2$-norm of Weight Vectors ($\sum_{k=1}^{K} \|\mathbf{w}_k\|_2^2$) be Small

Theorem 3 postulates that a decrease in the sum of the $l2$-norm of the weight matrix vectors for a GNN layer will result in a lower upper bound for classification error. Two commonly employed techniques to minimize the sum of the vector norm are batch normalization and L2-regularization. We carry out a thorough

analysis of the impacts of these two techniques on both the performance and the summation of vector norms, represented by $\sum_{k=1}^{K} \|\mathbf{w}_k\|_2^2$. The findings of our investigation are consolidated in Table 6 and 7. The results compellingly illustrate that batch normalization proves to be more effective in maintaining a relatively lower $\sum_{k=1}^{K} \|\mathbf{w}_k\|_2^2$ and concurrently improving performance. This comprehensive study underscores the critical role batch normalization plays in SP-GCL, thereby enhancing our understanding of SP-GCLs' operational nuances and guiding implementation efforts.

Table 6: The effect of BatchNorm within SP-GCL on the summation of $l2$-norm of weight matrix ( $\sum_{k=1}^{K} \|\mathbf{w}_k\|_2^2$) and performance. Comparing the network with same number of layers but w/wo batch normalization, the larger values of the summation of $l2$-norm of weight matrix are marked as red. And, the higher values of performance are marked are green.

| | Chameleon | | Squirrel | | Photo | | WikiCS | |
|---|---|---|---|---|---|---|---|---|
| | $l2$-norm sum. | Acc. | $l2$-norm sum. | Acc. | $l2$-norm sum. | Acc. | $l2$-norm sum. | Acc. |
| layer 1 w/ BN | [672.23] | 61.40 | [660.84] | 49.63 | [574.07] | 93.29 | [448.45] | 77.49 |
| layer 1 w/o BN | [686.11] | 41.49 | [676.79] | 33.87 | [609.15] | 26.68 | [449.51] | 24.68 |
| layer 2 w/ BN | [665.04, 513.83] | 63.33 | [658.96,516.03] | 51.91 | [568.14,520.50] | 94.06 | [442.24,515.64] | 78.36 |
| layer 2 w/o BN | [687.44,529.65] | 32.94 | [684.29,528.07] | 34.93 | [594.42,528.89] | 25.52 | [457.05,526.33] | 24.69 |
| layer 3 w/ BN | [663.36,513.50,514.62] | 61.60 | [657.24,515.71,513.84] | 51.40 | [566.54,519.24,520.13] | 93.57 | [441.72,515.12,515.25] | 78.27 |
| layer 3 w/o BN | [684.50,525.47,521.83] | 34.74 | [681.77,529.39,520.17] | 36.26 | [583.74,534.64,526.49] | 25.29 | [450.40,529.56,520.08] | 25.37 |

Table 7: The effect of $l2$ regularization on the summation of $l2$-norm of weight matrix ( $\sum_{k=1}^{K} \|\mathbf{w}_k\|_2^2$) and performance.

| | | $l2$ Regularization Strength | | | | |
|---|---|---|---|---|---|---|
| | | 1e-05 | 1e-04 | 1e-03 | 1e-02 | 1e-01 |
| Chameleon | Acc. | 41.49 | 41.45 | 41.40 | 41.43 | 41.51 |
| | $l2$ Norm. | 673.21 | 673.2 | 673.11 | 672.39 | 665.27 |
| Squirrel | Acc. | 27.87 | 27.89 | 27.88 | 27.88 | 27.83 |
| | $l2$ Norm. | 661.86 | 661.85 | 661.8 | 661.29 | 656.23 |
| Photo | Acc. | 26.32 | 26.56 | 26.78 | 26.86 | 25.87 |
| | $l2$ Norm. | 579.12 | 579.99 | 579.77 | 576.99 | 557.59 |
| WikiCS | Acc. | 24.68 | 24.68 | 24.66 | 24.67 | 24.55 |
| | $l2$ norm. | 448.28 | 448.44 | 448.36 | 447.62 | 439.54 |

## 6.4 Empirical Verification for the Theoretical Analysis

**Homophily of transformed graph (*Initial Stage*).** The homophily of transformed graph largely depends on the hidden representations. We theoretically shows the concentration property of the representations (Theorem 1) with mild assumptions. It implies that the transformed graph formed by selected positive pairs will have large edge homophily degree. But it is still unclear whether the derived conclusion is still hold over real-world datasets. Thus, we empirically measure the edge homophily of the transformed graph at the beginning of the training stage with 20 runs. The mean and standard deviation are reported in Table 5. The edge homophily of all transformed graphs are larger than the original one with a small standard deviation, except the CiteSeer in which a relatively high edge homophily (0.691) can still be achieved. The result indicates that the Theorem 1 is feasible over a wide-range real-world graph data.

**Distance to class center and Node Coverage (*Learning Process*).** We measure change of the average cosine distance ($1 - CosineSimilarity$) between the node embeddings and the class-center embeddings during training. Specifically, the class-center embedding is the average of the node embeddings of the same class. As shown in Figure 3, we found that the node embeddings will concentrate on their corresponding class centers during training, which implies that the learned embedding space becomes more compact and the learning process is stable. Intuitively, coupling with the "good" initialization as discussed above, the SP-GCL iteratively leverage the inductive bias to refine the embedding space. Furthermore, to study whether the representation of all nodes are benefited during the learning process, we measure the Node Cover Ratio and Overlapped Selection Ratio on four graph datasets. We denote the set of edges forming by the $K_{pos}$ positive selection at epoch $t$ as $e^t$ and define the Overlapped Selection Ratio as $\frac{e^{t+1} \cap e^t}{|e^{t+1}|}$. Besides, we denote the set of

positive nodes at at epoch $t$ as $v^t$. Then the Node Cover Ratio at epoch $t$ is defined as: $\frac{|v^0 \cap v^1 \cdots v^t|}{N}$. Following the hyperparameters described previously, we measure the node cover ratio and overlapped selection ratio during training. As shown in Figure 4, the overlapped selection ratio is low during the training and the node cover ratio coverage to 1 at the beginning of training. The result implies that all nodes can be benefited from the optimization procedure.

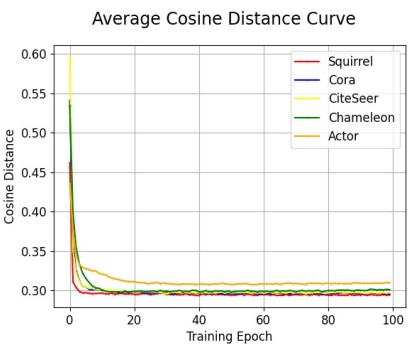

Figure 3: Average cosine distance between node embeddings and their corresponding class centers.

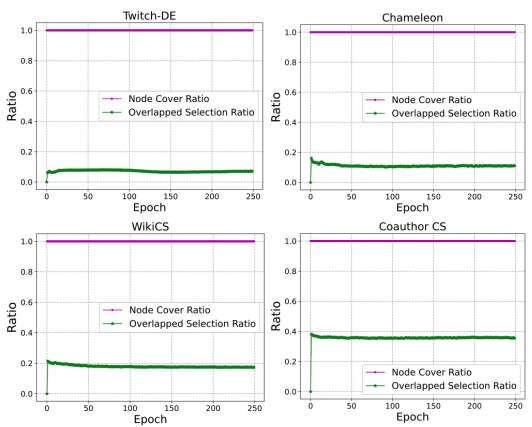

Figure 4: The Node Cover Ratio and Overlapped Selection Ratio.

**Selected positive pairs (*End of Training*).** To further study the effect of the proposed method, we provide the true positive ratio (TPR) of the selected pairs at the end of training with 20 runs. As shown in Table 4, the relatively large TPR and low variance suggests that the quality of the selected positive pairs is good and the learning process is stable.

## 6.5 Ablation Study for Hyper-parameter Selection

**Effect of Top-K Positive Sampling**. We study the effect of top-$K$ positive sampling on the node classification performance by measuring the classification accuracy and the corresponding standard deviation with a wide range of $K$. The results over WikiCS, Chameleon, Squirrel, Cora, and Photo are summarized in Figure 5. We observe that the performance achieved by SP-GCL is insensitive to the selection of $K$ from 2 to 18.

**Effect of $T$-hop Neighbors**. Factor $T$ is the other factor involved in node sampling. We provide a study about the the effect of $T$-hop neighbors on the node classification performance by measuring the classification accuracy or ROCAUC score and the corresponding standard deviation with different $T$. Specifically, with the same evaluation protocol described in Section A.3, we measure the classification accuracy on Squirrel, Cora, Actor, and Chameleon dataset and ROCAUC score on twitch-DE. As shown in Figure 5, the performance achieved by SP-GCL is insensitive to the selection of $T$.

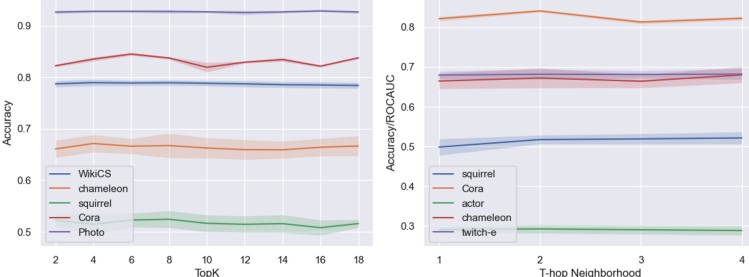

Figure 5: The effect of top-$K$ positive sampling and $T$-hop neighborhood on the performance.

# 7 Connection with Existing Observations

In Theorem 1, we theoretically analyze the concentration property of aggregated features of graphs following the Graph Assumption (Section 4.1). The concentration property has also been empirically observed in some other recent works (Wang et al., 2022b; Trivedi et al., 2022). Specifically, in Figure 1 of Wang et al. (2022b), a t-SNE visualization of node representations of the Amazon-Photo dataset shows that the representations obtained by a randomly initialized untrained SGC (Wu et al., 2019) model will concentrate to a certain area if they are from the same class. This observation also provides an explanation for the salient performance achieved by SP-GCL on the Amazon-Photo, as shown in Table 1.

# 8 Conclusion

In this work, to build a practical GCL approach that can perform well on both homophilic and heterophilic graphs, we begin by analyzing the concentration property of aggregated features for homophilic and heterophilic graphs. We then introduce the single-pass graph contrastive loss by utilizing the concentration property. We establish a link between the contrastive objective and the matrix factorization. Then, by utilizing the analysis of matrix factorization, we establish a theoretical guarantee for the downstream classification task using node representations learnt by optimizing the contrastive loss. We implement Single-Pass Graph Contrastive Learning (SP-GCL). Empirically, we demonstrate that SP-GCL outperforms or is competitive with SOTA approaches on 8 homophilic and 6 heterophilic graph benchmarks, while incurring a much lower computational overhead. The empirical findings demonstrate the viability and efficacy of our method in real-world situations.

# 9 Limitation and Future Work

Our work is only the first step in understanding the possibility of single-pass graph contrastive learning and the corresponding performance guarantee on both the homophilic graph and heterophilic graph. There is still a lot of future work to be done. Below we indicate three questions that need to be addressed.

- Our theoretical framework is built upon the graph assumption (Section 4.1) in which we assume the neighbor pattern. The graph assumption includes the homophilic graph and the "benign" heterophilic graph. However, there could exist graphs in which the neighbor pattern is messy or arbitrarily distributed. It is still an open question to understand the graph contrastive learning on those graphs.
- Recently, several Graph neural networks (Zhu et al., 2020b; Chien et al., 2020; Luan et al., 2021; Li et al., 2022) have been proposed to work on heterophilic graphs. Whether further improvement can be achieved through combining with those GNNs is still an open question.
- To verify the effectiveness of our theoretical findings, we keep the implementation to be simple. Whether a better empirical result can be achieved by involving more complex techniques still need to be explored.

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

# A Appendix: Experiment Setting

## A.1 Dataset Information

We analyze the quality of representations learned by SP-GCL on transductive node classification benchmarks. Specifically, we evaluate the performance of using the pretraining representations on 8 benchmark homophilic graph datasets, namely, Cora, Citeseer, Pubmed (Kipf & Welling, 2016a) and Wiki-CS, Amazon-Computers, Amazon-Photo, Coauthor-CS, Coauthor-Physics (Shchur et al., 2018), as well as 6 heterophilic graph datasets, namely, Chameleon, Squirrel (Rozemberczki et al., 2021), Actor (Pei et al., 2020), Twitch-DE, Twitch-gamers (Rozemberczki & Sarkar, 2021), and Genius (Lim et al., 2021b). The datasets are collected from real-world networks from different domains; their detailed statistics are summarized in Table 8. For the 8 homophilic graph data, we use the processed version provided by PyTorch Geometric (Fey & Lenssen, 2019). Besides, for the 6 heterophilic graph data, 3 of them, e.g., Chameleon, Squirrel and Actor are provided by PyTorch Geometric library. The other three dataset, genius, twitch-DE and twitch-gamers can be obtained from the official github repository[1], in which the standard splits for all the 6 heterophilic graph datasets can also be obtained. Those graph datasets follow the MIT license, and the personal identifiers are not included. We do not foresee any form of privacy issues.

Table 8: Statistics of datasets used in experiments.

| Name | Nodes | Edges | Class. | Feat. | $h_{node}$ | $h_{edge}$ | Avg. Degree |
|---|---|---|---|---|---|---|---|
| Cora | 2,708 | 5,429 | 7 | 1,433 | .825 | .810 | 4.01 |
| CiteSeer | 3,327 | 4,732 | 6 | 3,703 | .717 | .736 | 2.84 |
| PubMed | 19,717 | 44,338 | 3 | 500 | .792 | .802 | 4.50 |
| Coauthor CS | 18,333 | 327,576 | 15 | 6,805 | .832 | .808 | 35.74 |
| Coauthor Phy. | 34,493 | 991,848 | 5 | 8,451 | .915 | .931 | 57.51 |
| Amazon Comp. | 13,752 | 574,418 | 10 | 767 | .785 | .777 | 83.54 |
| Amazon Photo | 7,650 | 287,326 | 8 | 745 | .836 | .827 | 75.12 |
| WikiCS | 11,701 | 216,123 | 10 | 300 | .658 | .654 | 36.94 |
| Chameleon | 2,277 | 36,101 | 5 | 2,325 | .103 | .234 | 31.71 |
| Squirrel | 5,201 | 216,933 | 5 | 2,089 | .088 | .223 | 83.42 |
| Actor | 7,600 | 33,544 | 5 | 9,31 | .154 | .216 | 8.83 |
| Twitch-DE | 9,498 | 153,138 | 2 | 2,514 | .529 | .632 | 32.25 |
| Twitch-gamers | 168,114 | **6,797,557** | 2 | 7 | .552 | .545 | 80.86 |
| Genius | **421,961** | 984,979 | 2 | 12 | .477 | .618 | 4.67 |

## A.2 Baselines

We consider representative baseline methods belonging to the following three categories (1) Traditional unsupervised graph embedding methods, including DeepWalk (Perozzi et al., 2014), Node2Vec (Grover & Leskovec, 2016) and SELENE (Zhong et al., 2022), which fuses node attributes and network structure information without additional supervision. It employs a self-supervised learning loss on structural representations, and a reconstruction loss on node features. (2) Self-supervised learning algorithms with graph neural networks including Graph Autoencoders (GAE, VGAE) (Kipf & Welling, 2016b), Deep Graph Infomax (DGI) (Velickovic et al., 2019), Graphical Mutual Information Maximization (GMI) (Peng et al., 2020), and Multi-View Graph Representation Learning (MVGRL) (Hassani & Khasahmadi, 2020), graph contrastive representation learning (GRACE) (Zhu et al., 2020d) Graph Contrastive learning with Adaptive augmentation (GCA) (Zhu et al., 2021b), Bootstrapped Graph Latents (BGRL) (Thakoor et al., 2021), Augmentation-Free Graph Representation Learning (AFGRL) (Lee et al., 2021), (3) Supervised learning and Semi-supervised learning, e.g., Multilayer Perceptron (MLP) and Graph Convolutional Networks (GCN) (Kipf & Welling, 2016a), where they are trained in an end-to-end fashion.

## A.3 Evaluation Protocol

We follow the evaluation protocol of previous state-of-the-art graph contrastive learning approaches. Specifically, for every experiment, we employ the linear evaluation scheme as introduced in Velickovic et al. (2019), where each model is firstly trained in an unsupervised manner; then, the pretrained representations are used to train and test via a simple linear classifier. For the datasets that came with standard train/valid/test splits,

---

[1] https://github.com/CUAI/Non-Homophily-Large-Scale

we evaluate the models on the public splits. For datasets without standard split, e.g., Amazon-Computers, Amazon-Photo, Coauthor-CS, Coauthor-Physics, we randomly split the datasets, where 10%/10%/80% of nodes are selected for the training, validation, and test set, respectively. For most datasets, we report the averaged test accuracy and standard deviation over 10 runs of classification. While, following the previous works (Lim et al., 2021b;a), we report the test ROC AUC on genius and Twitch-DE datasets.

### A.4 Implementation Details

**Model architecture and hyperparamters.** We employ a two-layer GCN (Kipf & Welling, 2016a) as the encoder for all methods. The propagation for a single layer GCN is given by,

$$\text{GCN}_i(\mathbf{X}, \mathbf{A}) = \sigma\left(\bar{\mathbf{D}}^{-\frac{1}{2}}\bar{\mathbf{A}}\bar{\mathbf{D}}^{-\frac{1}{2}}\mathbf{X}\mathbf{W}_i\right),$$

where $\bar{\mathbf{A}} = \mathbf{A} + \mathbf{I}$ is the adjacency matrix with self-loops, $\bar{\mathbf{D}}$ is the degree matrix, $\sigma$ is a non-linear activation function, such as ReLU, and $\mathbf{W}_i$ is the learnable weight matrix for the $i$'th layer. Besides, following the previous works (Lim et al., 2021a;b), we use batch normalization without learnable affine parameters (to keep the implementation close to the theoretical analysis) within the graph encoder for the heterophilic graphs. The hyperparameters configuration for all experiments are summarized in Appendix A.

**Linear evaluation of embeddings.** In the linear evaluation protocol, the final evaluation is over representations obtained from pretrained model. When fitting the linear classifier on top of the frozen learned embeddings, the gradient will not flow back to the encoder. We optimize the one layer linear classifier 1000 epochs using Adam with learning rate 0.0005.

**Hardware and software infrastructures.** Our model are implemented with PyTorch Geometric 2.0.3 (Fey & Lenssen, 2019), PyTorch 1.9.0 (Paszke et al., 2017). We conduct experiments on a computer server with four NVIDIA Tesla V100 SXM2 GPUs (with 32GB memory each) and twelve Intel Xeon Gold 6240R 2.40GHz CPUs.

Table 9: Hyperparameter settings for all experiments.

|  | $lr.$ | $K_{pos}$ | $K_{neg}$ | $b$ | $T$ | $K$ |
|---|---|---|---|---|---|---|
| Cora | .0010 | 5 | 100 | 512 | 2 | 1024 |
| CiteSeer | .0020 | 10 | 200 | 256 | 2 | 1024 |
| PubMed | .0010 | 5 | 100 | 256 | 2 | 1024 |
| WikiCS | .0010 | 5 | 100 | 512 | 2 | 1024 |
| Amz-Comp. | .0010 | 5 | 100 | 256 | 2 | 1024 |
| Amz-Photo | .0010 | 5 | 100 | 512 | 2 | 1024 |
| Coauthor-CS | .0010 | 5 | 100 | 512 | 2 | 1024 |
| Coauthor-Phy. | .0010 | 5 | 100 | 256 | 2 | 1024 |
| Chameleon | .0010 | 5 | 100 | 512 | 3 | 1024 |
| Squirrel | .0010 | 5 | 100 | 512 | 3 | 1024 |
| Actor | .0010 | 10 | 100 | 512 | 4 | 1024 |
| Twitch-DE | .0010 | 5 | 100 | 512 | 4 | 1024 |
| Twitch-gamers | .0010 | 5 | 100 | 256 | 4 | 128 |
| Genius | .0010 | 5 | 100 | 256 | 3 | 512 |

## B More Experimental Results

### B.1 Single-Pass Augmentation-Free Graph Contrastive Learning with Different Types of Graph Encoders

In the previous section, we demonstrated the efficacy of the single-pass augmentation-free approach in Graph Contrastive Learning (GCL) using the Graph Convolutional Network (GCN) encoder (Kipf & Welling,

2016a). In the current section, we aim to further assess the adaptability of the proposed approach when paired with various types of graph encoders. Specifically, we elaborate on our experimental setup that incorporates three widely adopted graph encoders: Graph Isomorphism Network (GIN) (Xu et al., 2018), Graph Attention Network (GAT) (Veličković et al., 2017), and GraphSAGE (Hamilton et al., 2017). These experiments are conducted over a variety of graph datasets.

The result are summarized in Table 10. Our results underscore that our proposed contrastive learning technique exhibits consistent performance across a broad spectrum of graph neural networks. We posit that this consistency can be attributed to the shared foundational principle of these GNNs, specifically their reliance on neighborhood aggregation mechanisms. Furthermore, these results exemplify the flexibility and wide-ranging applicability of the SP-GCL approach, showcasing its potential to accommodate diverse graph encoder types and perform effectively across different graph data contexts.

| | Cora | WikiCS | Comp. | Amazon Photo | Phy. | Chameleon | Squirrel | Actor | Twitch-E |
|---|---|---|---|---|---|---|---|---|---|
| GIN | 79.45 | 72.73 | 85.14 | 90.31 | 93.88 | 60.22 | 40.88 | 27.65 | 69.70 |
| GIN+SP-GCL | 80.30 | 73.52 | 87.08 | 91.47 | 94.20 | 62.93 | 42.66 | 28.45 | 70.88 |
| GAT | 82.90 | 78.68 | 87.17 | 91.03 | 93.90 | 65.39 | 44.98 | 28.14 | 72.76 |
| GAT+SP-GCL | 83.11 | 79.96 | 89.52 | 93.22 | 96.05 | 65.99 | 46.56 | 30.70 | 73.43 |
| GraphSage | 81.70 | 76.06 | 86.72 | 92.08 | 93.14 | 64.61 | 43.70 | 32.06 | 70.60 |
| GraphSage+SP-GCL | 82.67 | 77.87 | 90.46 | 92.37 | 95.76 | 64.69 | 44.89 | 33.19 | 71.26 |

Table 10: The performance of SP-GCL combining with different graph neural networks as the encoder.

## C  Appendix: Detailed proofs

### C.1  Proof of Theorem 1

**Assumptions on graph data.** Consider the GNN model:

$$\mathbf{Z} = \sigma(\mathbf{D}^{-1}\mathbf{A}\mathbf{X}\ \mathbf{W}) \in \mathbb{R}^{N \times K}$$

where $\sigma$ represents the nonlinear activation function, the $\mathbf{X} \in \mathbb{R}^{N \times F}$ is the input layer, and $\mathbf{D} \in \mathbb{R}^{N \times N}$ is the degree matrix, $\mathbf{A} \in \mathbb{R}^{N \times N}$ is the adjacency matrix, and $\mathbf{W} \in \mathbb{R}^{F \times K}$ is the weight. Assume that $\|\mathbf{x}_i\|_2 \leq R$ for some $R \geq 0$, where $\mathbf{x}_i \in \mathbb{R}^F$ represents the node features of $i$-th input node (i.e., the transpose of the $i$-th row of $\mathbf{X}$). Define $\mathbf{w}_k \in \mathbb{R}^F$ to be the $k$-th column of $\mathbf{W}$. Given a label $y$, the node feature $\mathbf{x}$ is sampled from some distribution $P_{x|y}$. For node $v_i$, its neighbors labels are independently sampled from neighbor distribution $P_{y_i}$.

We assume that the nonlinear activation function $\sigma$ is homogeneous of degree $\tau$ for some $\tau \geq 1$ and $\sigma(q) \leq cq$ for some $c \geq 0$. For example, ReLU activation satisfies these assumptions with $\tau = c = 1$. Also, it recoverse the result of the linear model too since if $\sigma$ is identity (i.e., linear model), then $\tau = c = 1$.

**Theorem 1 (Concentration Property of Aggregated Features)** For any $\mathbf{W} \in \mathbb{R}^{F \times K}$, the following statements holds:

- For any pair $(i, i')$ such that $y_i = y_{i'}$,

$$\mathbb{E}[\mathbf{Z}_i] - \mathbb{E}[\mathbf{Z}_{i'}] = 0. \tag{9}$$

- With probability at least $1 - \delta$,

$$\|\mathbb{E}[\mathbf{Z}_i] - \mathbf{Z}_i\|_\infty \leq cR \left( \max_{k \in [K]} \|\mathbf{w}_k\|_2 \right) \sqrt{\frac{2\ln(2K/\delta)}{D_{ii}^{2\tau-1}}}. \tag{10}$$

- For any node pair $(i, j)$, with probability at least $1 - \delta'$,

$$|\mathbb{E}[\mathbf{Z}_i^\top \mathbf{Z}_j] - \mathbf{Z}_i^\top \mathbf{Z}_j| \leq c^2 R^2 \sum_{k=1}^{K} \|\mathbf{w}_k\|_2^2 \sqrt{\frac{2\ln(2/\delta')}{D_{ii}^{2\tau-1} D_{jj}^{2\tau-1}}}. \tag{11}$$

*Proof.* For any matrix $\mathbf{M}$, we use $\mathbf{M}_{i\cdot}$ and $\mathbf{M}_{\cdot j}$ to denote the $i$-th row vector and the $j$-th column vector of $\mathbf{M}$. We first calculate the expectation of aggregated embedding:

$$\mathbb{E}_X[\mathbf{Z}_i] = \mathbb{E}_X\left[\sigma\left(\sum_{j\in\mathcal{N}(i)}\frac{1}{D_{ii}}\mathbf{X}_{j\cdot}\mathbf{W}\right)\right] = \mathbb{E}_{(\mathbf{x}_j)_{j\in\mathcal{N}(i)}}\left[\sigma\left(\sum_{j\in\mathcal{N}(i)}\frac{1}{D_{ii}}\mathbf{x}_j^\top\mathbf{W}\right)\right]$$

$$= \mathbb{E}_{\bar{y}_k\sim P_{y_i},\bar{\mathbf{x}}_k\sim P_{x|\bar{y}_k},k=1,\ldots,D_{ii}}\left[\sigma\left(\sum_{j\in\mathcal{N}(i)}\frac{1}{D_{ii}}\bar{\mathbf{x}}_k^\top\mathbf{W}\right)\right]$$

Therefore, for any pair $(i, i')$ such that $y_i = y_{i'}$, we have

$$\mathbb{E}_X[\mathbf{Z}_i] - \mathbb{E}_X[\mathbf{Z}_{i'}] = 0.$$

Moreover,

$$
\begin{aligned}
\|\mathbb{E}_X[\mathbf{Z}_i] - \mathbf{Z}_i\|_\infty &= \left\|\mathbb{E}\left[\sigma\left(\sum_{j\in\mathcal{N}(i)}\frac{1}{D_{ii}}\mathbf{X}_{j\cdot}^\top\mathbf{W}\right)\right] - \sigma\left(\sum_{j\in\mathcal{N}(i)}\frac{1}{D_{ii}}\mathbf{X}_{j\cdot}^\top\mathbf{W}\right)\right\|_\infty \\
&= \left\|\frac{1}{D_{ii}^\tau}\left(\mathbb{E}\left[\sigma\left(\sum_{j\in\mathcal{N}(i)}\mathbf{X}_{j\cdot}^\top\mathbf{W}\right)\right] - \sigma\left(\sum_{j\in\mathcal{N}(i)}\mathbf{X}_{j\cdot}^\top\mathbf{W}\right)\right)\right\|_\infty \\
&= \max_{k\in[K]}\left\|\frac{1}{D_{ii}^\tau}\left(\mathbb{E}\left[\sigma\left(\sum_{j\in\mathcal{N}(i)}\mathbf{X}_{j\cdot}^\top\mathbf{W}_{\cdot k}\right)\right] - \sigma\left(\sum_{j\in\mathcal{N}(i)}\mathbf{X}_{j\cdot}^\top\mathbf{W}_{\cdot k}\right)\right)\right\|_\infty
\end{aligned}
\tag{12}
$$

where the first line follows the definition and the last line holds because the nonlinear activation function $\sigma$ is homogeneous of degree $\tau$. Furthermore, since $\sigma(q) \le cq$,

$$
\begin{aligned}
\left|\sigma\left(\sum_{j\in\mathcal{N}(i)}\mathbf{X}_{j\cdot}\mathbf{W}_{\cdot k}\right)\right| &\le c\left|\sum_{j\in\mathcal{N}(i)}\mathbf{X}_{j\cdot}\mathbf{W}_{\cdot k}\right| \le c\|\mathbf{W}_{\cdot k}\|_2\left\|\sum_{j\in\mathcal{N}(i)}\mathbf{X}_{j\cdot}\right\|_2 \\
&= c\|\mathbf{W}_{\cdot k}\|_2\sqrt{\sum_{j\in\mathcal{N}(i)}\sum_{t=1}^{F}X_{jt}^2} \\
&\le c\|\mathbf{W}_{\cdot k}\|_2\sqrt{\sum_{j\in\mathcal{N}(i)}R^2} \\
&= cR\|\mathbf{W}_{\cdot k}\|_2\sqrt{D_{ii}}
\end{aligned}
\tag{13}
$$

Thus, the random variable $v_k = \frac{1}{D_{ii}^\tau}\sigma\left(\sum_{j\in\mathcal{N}(i)}\mathbf{X}_{j\cdot}\mathbf{W}_{\cdot k}\right)$ is bounded as $-\frac{1}{D_{ii}^\tau}r \le v_k \le \frac{1}{D_{ii}^\tau}r$ where $r = cR\|\mathbf{W}_{\cdot k}\|_2\sqrt{D_{ii}}$. Thus, by applying Hoeffding's inequality to the RHS of equation (12),

$$\mathrm{P}\left(|v_k - \mathbb{E}[v_k]| \ge t\right) \le 2\exp\left(-\frac{t^2 D_{ii}^{2\tau}}{2r^2}\right)$$

Setting $\delta = 2\exp\left(-\frac{t^2 D_{ii}^{2\tau}}{2r^2}\right)$ and solving for $t$, this means that with probability at least $1 - \delta$,

$$|v_k - \mathbb{E}[v_k]| \le \sqrt{\frac{2r^2\ln(2/\delta)}{D_{ii}^{2\tau}}} = cR\|\mathbf{W}_{\cdot k}\|_2\sqrt{\frac{2D_{ii}\ln(2/\delta)}{D_{ii}^{2\tau}}} = cR\|\mathbf{W}_{\cdot k}\|_2\sqrt{\frac{2\ln(2/\delta)}{D_{ii}^{2\tau-1}}}.$$

By taking union bound over $k \in [K]$, with probability at least $1 - \delta$,

$$\|\mathbb{E}_X[\mathbf{Z}_i] - \mathbf{Z}_i\|_\infty = \max_{k\in[K]}|v_k - \mathbb{E}[v_k]| \le cR\left(\max_{k\in[K]}\|\mathbf{W}_{\cdot k}\|_2\right)\sqrt{\frac{2\ln(2K/\delta)}{D_{ii}^{2\tau-1}}}.$$

Furthermore,

$$|\mathbf{Z}_i^\top \mathbf{Z}_j| = \left|\sigma\left(\sum_{i'\in\mathcal{N}(i)} \frac{1}{D_{ii}}\mathbf{X}_{i'\cdot}\mathbf{W}\right)^\top \sigma\left(\sum_{j'\in\mathcal{N}(j)} \frac{1}{D_{jj}}\mathbf{X}_{j'\cdot}\mathbf{W}\right)\right| \tag{14}$$

$$= \left|\sum_k \sigma\left(\sum_{i'\in\mathcal{N}(i)} \frac{1}{D_{ii}}\mathbf{X}_{i'\cdot}\mathbf{W}_{\cdot k}\right)\cdot\sigma\left(\sum_{j'\in\mathcal{N}(j)} \frac{1}{D_{jj}}\mathbf{X}_{j'\cdot}\mathbf{W}_{\cdot k}\right)\right| \tag{15}$$

$$= \left(\frac{1}{D_{ii}^\tau}\frac{1}{D_{jj}^\tau}\right)\left|\sum_k \sigma\left(\sum_{i'\in\mathcal{N}(i)} \mathbf{X}_{i'\cdot}\mathbf{W}_{\cdot k}\right)\cdot\sigma\left(\sum_{j'\in\mathcal{N}(j)} \mathbf{X}_{j'\cdot}\mathbf{W}_{\cdot k}\right)\right| \tag{16}$$

$$\leq \left(\frac{1}{D_{ii}^\tau}\frac{1}{D_{jj}^\tau}\right)\sum_k \left|\sigma\left(\sum_{i'\in\mathcal{N}(i)} \mathbf{X}_{i'\cdot}\mathbf{W}_{\cdot k}\right)\right|\cdot\left|\sigma\left(\sum_{j'\in\mathcal{N}(j)} \mathbf{X}_{j'\cdot}\mathbf{W}_{\cdot k}\right)\right| \tag{17}$$

Coupling with the Equation (13),

$$|\mathbf{Z}_i^\top \mathbf{Z}_j| \leq \left(\frac{1}{D_{ii}^\tau}\frac{1}{D_{jj}^\tau}\right)\sum_k c^2 R^2 \|\mathbf{W}_{\cdot k}\|_2^2 \sqrt{D_{ii}D_{jj}}.$$

Thus, the random variable $s_{ij} = \left(\frac{1}{D_{ii}^\tau}\frac{1}{D_{jj}^\tau}\right)\left[\sum_k \sigma\left(\sum_{i'\in\mathcal{N}(i)} \mathbf{X}_{i'\cdot}\mathbf{W}_{\cdot k}\right)\cdot\sigma\left(\sum_{j'\in\mathcal{N}(j)} \mathbf{X}_{j'\cdot}\mathbf{W}_{\cdot k}\right)\right]$ is bounded as $-\left(\frac{1}{D_{ii}^\tau}\frac{1}{D_{jj}^\tau}\right)r \leq s_{ij} \leq \left(\frac{1}{D_{ii}^\tau}\frac{1}{D_{jj}^\tau}\right)r$ where $r = \sum_k \sigma\left(\sum_{i'\in\mathcal{N}(i)} \mathbf{X}_{i'\cdot}\mathbf{W}_{\cdot k}\right)\cdot\sigma\left(\sum_{j'\in\mathcal{N}(j)} \mathbf{X}_{j'\cdot}\mathbf{W}_{\cdot k}\right)$. Thus, by applying Hoeffding's inequality,

$$\mathrm{P}\left(|s_{ij} - \mathbb{E}[s_{ij}]| \geq t\right) \leq 2\exp\left(-\frac{t^2 D_{ii}^{2\tau} D_{jj}^{2\tau}}{2r^2}\right).$$

Correspondingly, we set $\delta' = 2\exp\left(-\frac{t^2 D_{ii}^{2\tau} D_{jj}^{2\tau}}{2r^2}\right)$ and solving for $t$, this means that with probability at least $1-\delta'$,

$$|s_{ij} - \mathbb{E}[s_{ij}]| \leq \sqrt{\frac{2r^2 \ln(2/\delta')}{D_{ii}^{2\tau} D_{jj}^{2\tau}}} = c^2 R^2 \sum_k \|\mathbf{W}_{\cdot k}\|_2^2 \sqrt{\frac{2\ln(2/\delta')}{D_{ii}^{2\tau-1} D_{jj}^{2\tau-1}}}.$$

With probability at least $1-\delta'$,

$$\left|\mathbb{E}_X[Z_i^\top Z_j] - Z_i^\top Z_j\right| \leq c^2 R^2 \sum_k \|\mathbf{W}_{\cdot k}\|_2^2 \sqrt{\frac{2\ln(2/\delta')}{D_{ii}^{2\tau-1} D_{jj}^{2\tau-1}}}.$$

## C.2 Proof of Lemma 1

To prove this lemma, we first introduce the concept of the probability adjacency matrix. For the transformed graph $\widehat{\mathcal{G}}$, we denote its probability adjacency matrix as $\widehat{\mathbf{W}}$, in which $\hat{w}_{ij} = \frac{1}{E}\cdot\widehat{\mathbf{A}}_{ij}$. $\hat{w}_{ij}$ can be understood as the probability that two nodes have an edge and the weights sum to 1 because the total probability mass is 1: $\sum_{i,j} \hat{w}_{i,j'} = 1$, for $v_i, v_j \in \mathcal{V}$. The corresponding symmetric normalized matrix is $\widehat{\mathbf{W}}_{sym} = \widehat{\mathbf{D}}_{\mathbf{w}}^{-1/2}\widehat{\mathbf{W}}\widehat{\mathbf{D}}_{\mathbf{w}}^{-1/2}$, and the $\widehat{\mathbf{D}}_{\mathbf{w}} = diag\left([\hat{w}_1,\ldots,\hat{w}_N]\right)$, where $\hat{w}_i = \sum_j \hat{w}_{ij}$. We then introduce the Matrix Factorization Loss which is defined as:

$$\min_{\mathbf{F}\in\mathbb{R}^{N\times k}} \mathcal{L}_{\mathrm{mf}}(\mathbf{F}) := \left\|\widehat{\mathbf{A}}_{sym} - \mathbf{F}\mathbf{F}^\top\right\|_F^2. \tag{18}$$

By the classical theory on low-rank approximation, Eckart-Young-Mirsky theorem (Eckart & Young, 1936), any minimizer $\widehat{\mathbf{F}}$ of $\mathcal{L}_{\mathrm{mf}}(\mathbf{F})$ contains scaling of the smallest eigenvectors of $\mathbf{L}_{sym}$ (also, the largest eigenvectors of $\widehat{\mathbf{A}}_{sym}$) up to a right transformation for some orthonormal matrix $\mathbf{R}\in\mathbb{R}^{K\times K}$. We have $\widehat{\mathbf{F}} = \mathbf{F}^*\cdot$ $\mathrm{diag}\left(\left[\sqrt{1-\lambda_1},\ldots,\sqrt{1-\lambda_K}\right]\right)\mathbf{R}$, where $\mathbf{F}^* = [\mathbf{u}_1, \mathbf{u}_2, \cdots, \mathbf{u}_K]\in\mathbb{R}^{N\times K}$. To proof the Lemma 1, we first present the Lemma 2.

**Lemma 2** *For transformed graph, its probability adjacency matrix $\widehat{\mathbf{W}}$, and adjacency matrix $\widehat{\mathbf{A}}$ are equal after the symmetric normalization, $\widehat{\mathbf{W}}_{sym} = \widehat{\mathbf{A}}_{sym}$.*

*Proof.* For any two nodes $v_i, v_j \in \mathcal{V}$ and $i \neq j$, we denote the the element in $i$-th row and $j$-th column of matrix $\widehat{\mathbf{W}}_{sym}$ as $\widehat{\mathbf{W}}_{sym}^{ij}$.

$$
\begin{aligned}
\widehat{\mathbf{W}}_{sym}^{ij} &= \frac{1}{\sqrt{\sum_k \hat{w}_{ik}} \sqrt{\sum_k \hat{w}_{kj}}} \frac{1}{E} \widehat{\mathbf{A}}^{ij} \\
&= \frac{1}{\sqrt{\sum_k \widehat{\mathbf{A}}_{ik}} \sqrt{\sum_k \widehat{\mathbf{A}}_{kj}}} \widehat{\mathbf{A}}^{ij} = \widehat{\mathbf{A}}_{sym}^{ij}
\end{aligned}
\tag{19}
$$

Leveraging the Lemma 2, we present the proof of Lemma 1.

**Proof of Lemma 1**  We start from the matrix factorization loss over $\widehat{\mathbf{A}}_{sym}$ to show the equivalence.

$$
\begin{aligned}
&\|\widehat{\mathbf{A}}_{sym} - \mathbf{F}\mathbf{F}^\top\|_F^2 = \|\widehat{\mathbf{W}}_{sym} - \mathbf{F}\mathbf{F}^\top\|_F^2 \\
&= \sum_{ij} \Big( \frac{\hat{w}_{ij}}{\sqrt{\hat{w}_i}\sqrt{\hat{w}_j}} - f_{\mathrm{mf}}(v_i)^\top f_{\mathrm{mf}}(v_j) \Big)^2 \\
&= \sum_{ij} (f_{\mathrm{mf}}(v_i)^\top f_{\mathrm{mf}}(v_j))^2 \\
&\quad - 2 \sum_{ij} \frac{\hat{w}_{ij}}{\sqrt{\hat{w}_i}\sqrt{\hat{w}_j}} f_{\mathrm{mf}}(v_i)^\top f_{\mathrm{mf}}(v_j) + \|\hat{\mathbf{W}}_{sym}\|_F^2 \\
&= \sum_{ij} \hat{w}_i \hat{w}_j \Big[ \Big( \frac{1}{\sqrt{\hat{w}_i}} \cdot f_{\mathrm{mf}}(v_i) \Big)^\top \Big( \frac{1}{\sqrt{\hat{w}_j}} \cdot f_{\mathrm{mf}}(v_j) \Big) \Big]^2 \\
&\quad - 2 \sum_{ij} \hat{w}_{ij} \Big( \frac{1}{\sqrt{\hat{w}_i}} \cdot f_{\mathrm{mf}}(v_i) \Big)^\top \Big( \frac{1}{\sqrt{\hat{w}_j}} \cdot f_{\mathrm{mf}}(v_j) \Big) + C
\end{aligned}
\tag{20}
$$

where $f_{\mathrm{mf}}(v_i)$ is the $i$-th row of the embedding matrix $\mathbf{F}$. The $\hat{w}_i$ which can be understood as the node selection probability which is proportional to the node degree. Then, we can define the corresponding sampling distribution as $P_{deg}$. If and only if $\sqrt{w_i} \cdot F_\psi(v_i) = f_{\mathrm{mf}}(v_i) = \mathbf{F}_i$, the we have:

$$
\begin{aligned}
&\mathbb{E}_{\substack{v_i \sim P_{deg} \\ v_j \sim P_{deg}}} \Big( F_\psi(v_i)^\top F_\psi(v_j) \Big)^2 \\
&- 2 \, \mathbb{E}_{\substack{v_i \sim Uni(\mathcal{V}) \\ v_{i+} \sim Uni(\mathcal{N}(v_i))}} \Big( F_\psi(v_i)^\top F_\psi(v_{i+}) \Big) + C
\end{aligned}
\tag{21}
$$

where $\mathcal{N}(v_i)$ denotes the neighbor set of node $v_i$ and $Uni(\cdot)$ is the uniform distribution over the given set. Because we constructed the transformed graph by selecting the top-$K_{pos}$ nodes for each node, then all nodes have the same degree. We can further simplify the objective as:

$$
\mathbb{E}_{\substack{v_i \sim Uni(\mathcal{V}) \\ v_j \sim Uni(\mathcal{V})}} \big( \mathbf{Z}_i^\top \mathbf{Z}_j \big)^2 - 2 \, \mathbb{E}_{\substack{v_i \sim Uni(\mathcal{V}) \\ v_{i+} \sim Uni(S_{pos}^i)}} \big( \mathbf{Z}_i^\top \mathbf{Z}_{i+} \big) + C.
\tag{22}
$$

Due to the node selection procedure, the factor $\sqrt{w_i}$ is a constant and can be absorbed by the neural network, $F_\psi$. Then, because $\mathbf{Z}_i = F_\psi(v_i)$, we can have the Equation 22. Therefore, the minimizer of matrix factorization loss is equivalent with the minimizer of the contrastive loss.

### C.3  Proof of Theorem 2

*Proof.* Now we provide the proof for the inner product of embedding. In particular, when we can achieve near-zero contrastive learning loss, *i.e.*, $\mathcal{L}_{\mathrm{mf}}(\mathbf{F}) = 0$, the minimizer $\mathbf{F}^*$ of $\mathcal{L}_{\mathrm{mf}}(\mathbf{F})$ contains scaling of the smallest eigenvectors of $\widehat{\mathbf{L}}_{sym}$ (also, the largest eigenvectors of $\widehat{\mathbf{A}}_{sym}$), $\mathbf{F}^* = [\mathbf{u}_1, \mathbf{u}_2, \cdots, \mathbf{u}_K] \in \mathbb{R}^{N \times K}$, according to the Eckart-Young-Mirsky theorem (Eckart & Young, 1936). Recall that $\mathbf{y} = \{1, -1\}^N \in \mathbb{R}^{N \times 1}$ is label of all nodes. Then we show there is a constraint on the quadratic form with respect to the optimal classifier. According to graph spectral theory (Chung & Graham, 1997), the quadratic form

$$
\mathbf{y}^\top \widehat{\mathbf{L}}_{sym} \mathbf{y} = \frac{1}{2} \sum_{i,i'} (\widehat{\mathbf{A}}_{sym})_{i,i'} (y_i - y_{i'})^2
$$

which captures the amount of edges connecting different labels. Furthermore, suppose that the expected homophily over distribution of graph feature and label, i.e., $y \sim P(y_i)$, $\mathbf{x} \sim P_y(\mathbf{x})$, through similarity selection satisfies $\mathbb{E}[h_{edge}(\hat{\mathcal{G}})] = 1 - \bar{\phi}$. Here $\bar{\phi} = \mathbb{E}_{v_i, v_j \sim Uni(\mathcal{V})}(\hat{\mathbf{A}}_{i,j} \cdot \mathbb{1}[y_i \neq y_j])$. Since we have defined that $\bar{\phi}$ as the density of edges that connects different labels, we can directly show that $\mathbf{y}^\top \hat{\mathbf{L}}_{sym} \mathbf{y} \leq \bar{\phi} N$. Then we expand the above expression accoding to $\hat{\mathbf{L}}_{sym} = \mathbf{I} - \hat{\mathbf{A}}_{sym}$,

$$\mathbf{y}^\top \hat{\mathbf{L}}_{sym} \mathbf{y} = \mathbf{y}^\top (\mathbf{I} - \hat{\mathbf{A}}_{sym}) \mathbf{y} = N - \mathbf{y}^\top \hat{\mathbf{A}}_{sym} \mathbf{y}$$

Next we consider the situation that we achieve the optimal solution of the contrastive loss, $\mathbf{Z} = \arg\min \mathcal{L}_{\text{SP-GCL}}$. Then, we have $\mathcal{L}_{mf}(\mathbf{F}^*) = 0$, which implies that $\hat{\mathbf{A}}_{sym} = (\mathbf{F}^*)^\top \mathbf{F}^*$. As we analyzed in Section C.2, we have $\mathbf{Z} = \mathbf{F}^*$ in this case. Furthermore, we have,

$$\begin{aligned}
\mathbf{y}^\top \hat{\mathbf{L}}_{sym} \mathbf{y} &= \mathbf{y}^\top (\mathbf{I} - \hat{\mathbf{A}}_{sym}) \mathbf{y} = N - \mathbf{y}^\top \mathbf{Z}^\top \mathbf{Z} \mathbf{y} \\
&= N - \mathbf{y}^\top (\mathbf{Z}_i^\top \mathbf{Z}_j)_{(i,j \in n \times n)} \mathbf{y} \\
&= N - \Big( \sum_{i,j} \mathbf{Z}_i^\top \mathbf{Z}_j |_{y_i = y_j} - \sum_{i,j} \mathbf{Z}_i^\top \mathbf{Z}_j |_{y_i \neq y_j} \Big)
\end{aligned} \tag{23}$$

This leads to,

$$\begin{aligned}
&\frac{1}{N} \Big[ \sum_{i,j} \mathbf{Z}_i^\top \mathbf{Z}_j |_{y_i = y_j} - \sum_{i,j} \mathbf{Z}_i^\top \mathbf{Z}_j |_{y_i \neq y_j} \Big] \\
&= \mathbb{E}[\mathbf{Z}_i^\top \mathbf{Z}_j |_{y_i = y_j}] - \mathbf{E}[\mathbf{Z}_i^\top \mathbf{Z}_j |_{y_i \neq y_j}] \geq 1 - \bar{\phi}
\end{aligned} \tag{24}$$

### C.4 Proof of Theorem 3

Recently, HaoChen et al. (2021) presented the following theoretical guarantee for the model learned with the matrix factorization loss.

**Lemma 3** *For a graph $\mathcal{G}$, let $f_{\text{mf}}^* \in \arg\min_{f_{\text{mf}}: \mathcal{V} \to \mathbb{R}^K}$ be a minimizer of the matrix factorization loss, $\mathcal{L}_{\text{mf}}(\mathbf{F})$, where $\mathbf{F}_i = f_{\text{mf}}(v_i)$. Then, for any label $\mathbf{y}$, there exists a linear classifier $\mathbf{B}^* \in \mathbb{R}^{c \times K}$ with norm $\|\mathbf{B}^*\|_F \leq 1/(1 - \lambda_K)$ such that*

$$\mathbb{E}_{v_i} \left[ \|\vec{y}_i - \mathbf{B}^* f_{\text{mf}}^*(v_i)\|_2^2 \right] \leq \frac{\phi^{\mathbf{y}}}{\lambda_{K+1}}, \tag{25}$$

*where $\vec{y}_i$ is the one-hot embedding of the label of node $v_i$. The difference between labels of connected data points is measured by $\phi^{\mathbf{y}}$, $\phi^{\mathbf{y}} := \frac{1}{E} \sum_{v_i, v_j \in \mathcal{V}} \mathbf{A}_{ij} \cdot \mathbb{1}[y_i \neq y_j]$.*

*Proof of Theorem 3.* This proof is a direct summary on the established lemmas in previous section. By Lemma 1 and Lemma 3, we have,

$$\mathbb{E}_{v_i} \left[ \|\vec{y}_i - \mathbf{B}^{*\top} \mathbf{Z}_i^*\|_2^2 \right] \leq \frac{\phi^y}{\hat{\lambda}_{K+1}} \tag{26}$$

where $\hat{\lambda}_i$ is the $i$-th smallest eigenvalue of the Laplacian matrix $\hat{\mathbf{L}}_{sym} = \mathbf{I} - \hat{\mathbf{A}}_{sym}$. $\mathbf{Z}^*$ is obtained through the non-linear graph neural network, and the nonlinearity follows the assumption 2. Note that $\phi^y$ of Lemma 3 equals $1 - h_{edge}$.

Then we apply Theorem 1 and Theorem 2 for $h_{edge}$ to conclude the proof:

$$
\begin{aligned}
&\mathbb{E}_{v_i}\left[\left\|\vec{y}_i - \mathbf{B}^{*\top}\mathbf{Z}_i^*\right\|_2^2\right]\\
&\leq \frac{1 - h_{edge}}{\hat{\lambda}_{K+1}} \leq \frac{\bar{\phi} + \mathbb{E}_{v_i,v_j\sim Uni(\mathcal{V})}c^2R^2\left[\sum_{k=1}^{K}\|\mathbf{w}_k\|_2^2\sqrt{\frac{2\ln(2/\delta')}{D_{ii}^{2\tau-1}D_{jj}^{2\tau-1}}}\right]}{\hat{\lambda}_{K+1}}\\
&= \frac{\bar{\phi}}{\hat{\lambda}_{K+1}} + c^2R^2\sum_{k=1}^{K}\|\mathbf{w}_k\|_2^2\sqrt{\frac{2\ln(2/\delta')}{\hat{\lambda}_{K+1}^2}}\mathbb{E}_{v_i,v_j\sim Uni(\mathcal{V})}\left[(D_{ii}D_{jj})^{-\frac{1}{2}(2\tau-1)}\right]\\
&\leq \frac{\bar{\phi}}{\hat{\lambda}_{K+1}} + c^2R^2\sum_{k=1}^{K}\|\mathbf{w}_k\|_2^2\sqrt{\frac{2\ln(2/\delta')}{\hat{\lambda}_{K+1}^2}}\mathbb{E}_{v_i\sim Uni(\mathcal{V})}\left[(D_{ii})^{-(2\tau-1)}\right]\\
&= \frac{\bar{\phi}}{\hat{\lambda}_{K+1}} + c^2R^2 C_{degree}\sum_{k=1}^{K}\|\mathbf{w}_k\|_2^2\sqrt{\frac{2\ln(2/\delta')}{\hat{\lambda}_{K+1}^2}}
\end{aligned}
\tag{27}
$$

where $C_{degree} = \mathbb{E}_{v_i\sim Uni(\mathcal{V})}\left[(D_{ii})^{-(2\tau-1)}\right] > 0$ is determined by input graph structure and is invariant to the learning process. The positive real constant $R$ and $c$ are defined in Assumption 1 and 2 respectively.

## D   Augmentations in Existing Works

Graph augmentation techniques are commonly used in Graph Contrastive Learning (GCL) to generate different views of the same graph, facilitating the learning of invariant and robust representations. The following sections provide detailed insights into the methods mentioned in Table 11, including edge removing, edge adding, node dropping, and subgraph induction by random walks.

Edge Removing: This is a simple yet effective augmentation technique where random edges are removed from the graph to create a new view. The goal is to train the model to learn representations that can predict the missing edges, enhancing its ability to capture the underlying connectivity pattern of the graph. This method requires careful calibration as removing too many edges might disrupt the graph structure, whereas removing too few may not provide enough learning signal for the model.

Edge Adding: Complementary to edge removing, edge adding involves introducing new edges to the graph. The new edges can be randomly generated or based on certain heuristics such as node similarity. The objective here is to encourage the model to recognize and adapt to changes in the graph's connectivity. This augmentation can help the model understand the importance of each edge and improve its ability to generalize to unseen graph structures.

Node Dropping: In this augmentation technique, random nodes (along with their associated edges) are dropped from the graph. This can help the model learn robust representations that are resilient to changes in the node set. However, similar to edge removing, the number of nodes to be dropped needs to be carefully tuned to maintain the balance between providing enough learning signal and preserving the essential structure of the graph.

Subgraph Induced by Random Walks: This method involves generating a subgraph by performing random walks on the original graph. Each random walk starts from a node and moves to its neighboring nodes randomly, creating a path. The subgraph consists of the nodes and edges visited during the walk. This technique can reveal local neighborhood structures and dependencies among nodes, aiding the model in learning local as well as global graph structures.

Node Shuffling: Node shuffling is a technique where the identities or features of nodes in the graph are permuted, meaning that the feature vector of a node is reassigned to another node. This is done while maintaining the adjacency structure of the graph, essentially changing the correspondence between nodes and their features without changing the graph topology. This can help the model to learn representations that are invariant to changes in node identities or features.

Diffusion: Diffusion is a process that propagates information (or features) across the graph based on the adjacency matrix of the graph. For graph augmentation, a diffusion process can be used to create a new graph where node features are updated based on their neighbors' features. This can simulate the spread of information or influence in the graph, and the resulting graph can be used as a different view for contrastive learning.

Table 11: Summary of graph augmentations used by representative GCL models. Multiple$^*$ denotes multiple augmentation methods including edge removing, edge adding, node dropping and subgraph induced by random walks.

| Method | Topology Aug. | Feature Aug. |
|---|---|---|
| DGI (Velickovic et al., 2019) | Node Shuffling | - |
| GMI (Peng et al., 2020) | Node Shuffling | - |
| MVGRL (Hassani & Khasahmadi, 2020) | Diffusion | - |
| GCC (Qiu et al., 2020) | Subgraph | - |
| GraphCL (You et al., 2020) | Multiple$^*$ | Feature Dropout |
| GRACE (Zhu et al., 2020d) | Edge Removing | Feature Masking |
| GCA (Zhu et al., 2021b) | Edge Removing | Feature Masking |
| BGRL (Grill et al., 2020) | Edge Removing | Feature Masking |

