# OpenReview forum: "Single-Pass Contrastive Learning Can Work for Both Homophilic and Heterophilic Graph"
_TMLR — Accepted by TMLR_

### Review · Reviewer_31ov · 2023-05-02

**Summary Of Contributions:**

This paper studies Graph Contrastive Learning (GCL) on both homophilic and heterophilic graphs. The authors propose a concentration property for features obtained by neighborhood aggregation on homophilic and heterophilic graphs, and find that the inner product of hidden representations approximates to its class-wise expectation with a high probability, no matter on homophilic or heterophilic graphs. Then, the authors leverage the property to sample positive and negative samples based the hidden representation similarity, and developed a new GCL algorithm called Single-Pass GCL (SP-GCL), which does not need multiple passes as required by previous augmentation based GCL approaches. The authors conduct extensive experiments on multiple homophilic and heterophilic benchmarks and verify the effectiveness of SP-GCL.



**Audience:**

Yes

**Claims And Evidence:**

No

**Requested Changes:**

While I acknowledge the considerable effort put into the experiments, I believe there is room for improvement in the work, specifically in addressing the following weaknesses:

i) (as well as iv) The motivation of the work is not clearly presented: To my understanding, the authors seem to claim that previous dual-pass design does not have theoretical guarantees and can be costly on large-scale graphs. However, there has already been formal analysis of GCL generalization ability in [1], as well as efficient design for dual-pass algorithms in [2].

Addressing the heterophilic node classification via GCL is the main motivation of this work, while the node feature based homophily has already been proposed in [3]. If so, why not directly using the augmentation based strategies, but instead using the single pass sampling?

I believe the motivation of this work could be presented more clearly by clarifying the aforementioned issues as well as complementing the related work discussion.


ii) The concentration property theory poses strong assumptions and is limited to a specific graph family:
- The node feature $x$ is independently sampled from $P_{x|y}$ solely based on the label $y$, which does not consider the graph structures. It seems merely using a MLP is sufficient for learning the desired node representation.
- The neighbors’ labels are further sampled from $P_y$, but neither $P_y$ nor $P_{x|y}$ are formally defined. Why do the authors adopt such a graph data model? How can the data model reflect the graphs generation in real-world applications?
- It’s unclear why “the assumption means that the label of a neighbor is only determined by the label of the central node”, and why it is feasible for both homophily and heterophily.


iii) The empirical improvements are limited:
- On homophilic graphs, previous GCL approaches seem to perform better than SP-GCL;
- On heterophilic graphs, SP-GCL seems to perform better than previous GCL methods on small scale datasets while the improvements are decreased for large scale datasets;
- SP-GCL can not outperform the MLP on heterophilic graphs, which seems to justify the limitations of the theoretical assumptions.

Besides, only GCN is considered as the backbone in the experiments. Is SP-GCL also compatible with other more complicated backbones such as Sage, GIN, and GAT? Can the analysis in Section 6.7 also hold for those complicated backbones?

The computational cost analysis seems not consider the transformation operations in SP-GCL.

**Minor**
- “, And the node homophily” in Section 3 should be “. And”;
- The font size for “SP-GCL” in Table 4 seems to be too small;


**References**

[1] Improving graph representation learning by contrastive regularization, arXiv 2021.

[2] Rethinking and Scaling Up Graph Contrastive Learning: An Extremely Efficient Approach with Group Discrimination, NeurIPS 2022.

[3] Understanding and Improving Graph Injection Attack by Promoting Unnoticeability, ICLR 2022.


**Strengths And Weaknesses:**

**Strengths**

i) The concentration property seems to be new and interesting, which provides a homophily agnostic characterization of node representations in GCL.

ii) The experiments are extensive and cover a broad scope of homophilic and heterophilic datasets, with various homophily degrees. The improvements of SP-GCL over previous GCL approaches on multiple heterophilic datasets are impressive.

iii) The authors also provide plentiful analysis to demonstrate the usefulness of the proposed sampling (or graph transformation) method, as well as the applicability of theoretical assumptions.

**Weakness**

i) The motivation of the work is not clearly presented.

ii) The concentration property theory assumes a specific graph family, which could rarely hold in practice.

iii) The experimental improvements are limited on various datasets. The experiments merely consider a specific GNN backbone, i.e., GCN. It’s unknown whether SP-GCL can be applied to more complicated GNN backbones, such as GAT, Sage and GIN.

iv) Some related works are missing discussion.

---

> ### Author Response · Authors · 2023-06-01
> **Reply for Reviewer 31ov (I)**
>
> Thank you for the comments in the "Requested Changes" section. Before we address the concerns you've highlighted, we'd like to re-emphasize our theoretical contributions. Our implementation directly builds upon the results of our analysis, devoid of any specifically designed techniques intended to heuristically boost performance. The experiments act as empirical justifications supporting this theoretical analysis. To better demonstrate the novelty and practicality of our key theorem (Theorem 3), we have included a remark section in our revision dedicated to discussing it. Additionally, we've added another section to validate the usefulness of the insights obtained from Theorem 3, in shaping the design of our model.
>
> **Q1.1 Exisiting works[1,2] hurt the claim of this work.**
> > First, This work addresses limitations of existing Graph Contrastive Learning (GCL) techniques in handling heterophilic graphs and providing robust performance guarantees. The goal is to develop a new GCL method, Single-Pass Graph Contrastive Learning (SP-GCL), that is applicable to both homophilic and heterophilic graphs and comes with performance assurances.
> "Previous dual-pass design can be costly on large-scale graphs" is not the claim of the paper. The efficiency of previous dual-pass design is not even discussed in both abstract and introduction section. Therefore, the preivous work [2] foucsing on efficiency of GCLs doesn't hurt the motivation of this work.
> The efficient is a byproduct of our single-pass agumentation-free design. Besides the efficiency study of the experiment section aims to show the proposed single-pass agumentation-free design itself is efficient enough for large graph.
>
> > The paper [1] is not a analysis for GCL generalization. The Section 4 of the paper [1] provide an analysis for NCEloss used in [4,5,6]. In our method, we didn't use the NCEloss. The analysis of paper [1] is neither about GCL nor the loss proposed in this work. Therefore,  we don't think it will hinge the motivation of this work.
>
>
> **Q1.2 Due to the  existence of work[3], why not directly using the augmentation based strategies, but instead using the single pass sampling?**
>
> > The research "Understanding and Improving Graph Injection Attack by Promoting Unnoticeability"[3] presents an innovative approach to Graph Injection Attack (GIA), emphasizing the preservation of graph homophily. This methodology enhances the difficulty of defending against the attack while inducing considerable performance decay in Graph Neural Networks (GNNs). The authors introduce the notion of homophily unnoticeability and the Harmonious Adversarial Objective (HAO) to accomplish these aims. However, our study distinguishes itself by focusing on graph contrastive learning for both homophilic and heterophilic graphs, specifically targeting node classification tasks. While there may be potential to apply insights from paper [3] to devise a method suitable for both graph types, it's crucial to recognize the existing divergence between our work and theirs. The creation of such a method would necessitate substantial effort and meticulous planning, constituting a non-trivial task worthy of future investigation. Implementing the augmentation-based strategies from paper [3] directly may be feasible, but it's beyond the scope of our current discussion and doesn't diminish the motivation or contributions of our present work.

---

> > ### Author Response · Authors · 2023-06-01
> > **Reply for Reviewer 31ov (II)**
> >
> > **Q2.1 Independently sampled node feature without considering the graph structures is infeasible and make a MLP is sufficient for learning the desired node representation.**
> >
> > > To clarify the hypothesis regarding graph data, we have distinctly articulated it as Assumption 1 in our revision. Notably, this graph data assumption (Assumption 1 in our revision) is a widely accepted presupposition for theoretical analysis in the graph domain, as evidenced by references [7, 8, 9, 10]. It's not just prevalent in theoretical analysis, but is also generally believed that the insights and methods derived based on it are practical for real-world scenarios, as indicated by references [11,12].
> >
> > > Secondly, assuming that node features are independently sampled from a distribution doesn't necessarily mean that a Multilayer Perceptron (MLP) is adequate. As an illustration, let's assume two classes of nodes are independently and identically distributed, sampled from two Gaussian distributions (disregarding the graph structures). There is a level of overlap between these two classes' features in high-dimensional space. Consequently, an MLP may not be sufficient in this case. Similar discussions can be found in previous works [7,8].
> >
> > > Lastly, the assumption of generating node features without considering the graph structures is a reasonable one. For instance, in real-world scenarios like citation networks, node features originate from the content of titles and abstracts, while the graph structure results from citations. It's clear that the generation of these features doesn't take the graph structure into account.
> >
> >
> >
> > **Q2.2 The neighbors' labels are further sampled from $P_y$, but neither $P_y$ nor $P_{x|y}$  are formally defined. Why do the authors adopt such a graph data model? How can the data model reflect the graphs generation in real-world applications?**
> >
> > > Thank you for your suggestions. In response, we've explicitly articulated the definitions of $P_y$ and $P_{x|y}$ in Assumption 1 of our revision, accompanied by a pertinent remark (Note, the $P_y$ and $P_{x|y}$ cannot be explicitly defined). This graph data model has been selected due to its widespread use as a fundamental assumption in graph data studies, as indicated by references [7,8,9,10].
> >
> > > Moreover, it's crucial to underscore that this model isn't just theoretically appealing, but its practical relevance has also been substantiated. The theoretical outcomes derived from this data model have been extensively validated as practical, echoing our previous discussion. It is noteworthy that this model accurately embodies the generation of graphs in real-world applications, where nodes and their features often exhibit dependence, while preserving an inherent structure within the data. This makes it a fitting choice for studies that intend to uncover or leverage these inherent patterns.
> >
> > > In real-world scenarios, nodes often represent entities whose features are determined by their individual characteristics and the roles they play within their networks. This dependency is elegantly captured by our data model, which presumes that node features, captured by $P_{x|y}$, are dependent on the node labels, represented by $P_y$. Thus, our graph data model aligns well with many real-world graph generation mechanisms, providing a solid basis for our study and its potential applications.

---

> > > ### Author Response · Authors · 2023-06-01
> > > **Reply for Reviewer 31ov (III)**
> > >
> > > **Q2.3 why “the assumption means that the label of a neighbor is only determined by the label of the central node”, and why it is feasible for both homophily and heterophily.**
> > >
> > > > We appreciate your question. In our model, we've made the assumption that for a given node, $v_i$, the labels of its neighboring nodes are independently sampled from a neighbor distribution, $P_{y_i}$. This implies that the label of a neighbor node is determined solely by the label of the central node it is associated with. Essentially, the labeling mechanism is driven by the label of the central node, which makes the label of the neighbor node contingent upon it.
> > >
> > > > Homophily and heterophily are two distinct principles that describe the nature of node connections in a graph. Homophily corresponds to the tendency of nodes to form links with others that share similar features or labels. In contrast, heterophily represents situations where nodes are more likely to connect with others that are different in features or labels.
> > >
> > > > When considering these principles in the context of our assumption, if $P_{y_i}$ is biased such that it has a high probability of sampling a label that matches $y_i$, the central node's label, then we observe a trend towards homophily. This means that nodes are more likely to link with nodes of the same label, reflecting a homophilic graph.
> > >
> > > > Conversely, if the probability distribution $P_{y_i}$ tends to sample labels that are different from $y_i$, the graph displays a tendency towards heterophily. This is because nodes are more likely to form connections with nodes that have different labels, echoing the definition of a heterophilic graph.
> > >
> > > > This mechanism makes our assumption versatile as it can feasibly model both homophilic and heterophilic scenarios by simply adjusting the underlying neighbor distribution, $P_{y_i}$, from which we sample the neighboring node labels.
> > >
> > > **Q3.1 On homophilic graphs, previous GCL approaches seem to perform better than SP-GCL.**
> > > > In fact, our results suggest otherwise. SP-GCL demonstrates superior performance compared to all previous GCL methodologies when applied to homophilic graphs. We encourage you to review Table 1 for a more detailed comparison. We have recently updated this table to include the average scores, further highlighting the advantageous performance of SP-GCL. This improved performance underscores the advancements made with SP-GCL in handling homophilic graphs.
> > >
> > >
> > > **Q3.2 On heterophilic graphs, SP-GCL seems to perform better than previous GCL methods on small scale datasets while the improvements are decreased for large scale datasets.**
> > > > Contrary to the impression, our observations suggest that SP-GCL's performance on heterophilic graphs is not only superior to previous GCL methods on small scale datasets, but the advantage is more pronounced for larger scale datasets such as Twitch-gamers and Genius. This trend contradicts the notion of decreased improvements with larger datasets. We invite you to refer to Table 2, which provides a comprehensive view of this performance differentiation across various dataset sizes. These results reaffirm the scalability and effectiveness of SP-GCL, particularly with regard to handling heterophilic graphs of larger dimensions.
> > >
> > >
> > > **Q3.3 SP-GCL can not outperform the MLP on heterophilic graphs, which seems to justify the limitations of the theoretical assumptions.**
> > > > Firstly, On average, SP-GCL exhibits superior performance over MLP on heterophilic graphs (Table 2). While it's true that MLP may yield superior results on certain heterophilic graphs, this does not necessarily imply that the theoretical assumptions underlying SP-GCL are limited. It's essential to note that our theoretical assumptions, and the corresponding theoretical analysis, are intended for unsupervised learning scenarios. Therefore, the comparison should not be directly drawn between the accuracy bounds of supervised training and those of contrastive learning.
> > >
> > > > The purpose of our theory is to highlight the potential of unsupervised learning and its possible applications in various situations, including heterophilic graphs. It's not to claim supremacy over all learning methods under all circumstances. Thus, the relative performance of MLP on specific datasets does not inherently constrain the utility or accuracy of our theoretical model, as they target different learning paradigms.
> > >
> > > > Therefore, while we acknowledge that SP-GCL might not outperform MLP in some heterophilic graph scenarios, we maintain that our theoretical assumptions remain valid and robust in the domain they are intended to address.

---

> > > > ### Author Response · Authors · 2023-06-01
> > > > **Reply for Reviewer 31ov (IV)**
> > > >
> > > > **Q4 SP-GCL is compatible with other more complicated backbones. Can the analysis in Section 6.7 also hold for those complicated backbones?**
> > > > > Yes, SP-GCL is indeed compatible with a variety of more intricate backbone architectures. We illustrate this in the Appendix, specifically in Section B.1, where we conduct an empirical study that involves substituting the backbone with other established models such as GAT, GIN, and GraphSAGE. Upon analyzing diverse datasets, we found that the performance delivered by SP-GCL consistently outperformed those of its supervised counterparts, demonstrating the method's adaptability.
> > > >
> > > > > However, it's necessary to emphasize that the analysis detailed in Section 6.7 doesn't straightforwardly extend to all other graph neural networks. This stipulation arises because we've expressly defined ${\mathbf{Z}} =\sigma(\textbf{D}^{-1}\textbf{AX W}) \in \mathbb{R}^{ N \times K}$ in Section 4.1. Nonetheless, it's entirely possible to obtain similar results if we modify the definition of ${\mathbf{Z}}$ and repeat the associated derivation process. This re-derivation for various backbone architectures, while feasible, falls outside the current scope of our work.
> > > >
> > > > > In conclusion, SP-GCL exhibits compatibility with other complex GNN backbones, and with some modifications and re-derivations, the analytical framework can indeed be extended to accommodate these backbones. This underscores the versatility and adaptability of our proposed method, SP-GCL.
> > > >
> > > >
> > > > **Q5 computational cost analysis seems not consider the transformation operations in SP-GCL.**
> > > > > The transformation operation in SP-GCL has indeed been taken into account. The term associated with $C_{prod}$ specifically represents the computational cost for these transformation operations. This ensures that all key components of the SP-GCL are incorporated in our analysis, thus providing a comprehensive view of the overall computational expense.
> > > >
> > > >
> > > >
> > > > Reference:
> > > >
> > > > [1] Ma, Kaili, Haochen Yang, Han Yang, Tatiana Jin, Pengfei Chen, Yongqiang Chen, Barakeel Fanseu Kamhoua, and James Cheng. "Improving graph representation learning by contrastive regularization." arXiv preprint arXiv:2101.11525 (2021).
> > > >
> > > > [2] Zheng, Yizhen, Shirui Pan, Vincent Lee, Yu Zheng, and Philip S. Yu. "Rethinking and scaling up graph contrastive learning: An extremely efficient approach with group discrimination." Advances in Neural Information Processing Systems 35 (2022): 10809-10820.
> > > >
> > > > [3] Chen, Yongqiang, Han Yang, Yonggang Zhang, Kaili Ma, Tongliang Liu, Bo Han, and James Cheng. "Understanding and improving graph injection attack by promoting unnoticeability." arXiv preprint arXiv:2202.08057 (2022).
> > > >
> > > > [4] Belghazi, Mohamed Ishmael, Sai Rajeswar, Aristide Baratin, Devon Hjelm, and Aaron Courville. "MINE: Mutual Information Neural Estimation."
> > > >
> > > > [5] Chen, Ting, Simon Kornblith, Mohammad Norouzi, and Geoffrey Hinton. "A simple framework for contrastive learning of visual representations." In International conference on machine learning, pp. 1597-1607. PMLR, 2020.
> > > >
> > > > [6] Henaff, Olivier. "Data-efficient image recognition with contrastive predictive coding." In International conference on machine learning, pp. 4182-4192. PMLR, 2020.
> > > >
> > > > [7] Baranwal, Aseem, Kimon Fountoulakis, and Aukosh Jagannath. "Effects of graph convolutions in deep networks." arXiv preprint arXiv:2204.09297 (2022).
> > > >
> > > > [8] Baranwal, Aseem, Kimon Fountoulakis, and Aukosh Jagannath. "Graph convolution for semi-supervised classification: Improved linear separability and out-of-distribution generalization." arXiv preprint arXiv:2102.06966 (2021).
> > > >
> > > > [9] Yang, Shenghao, and Kimon Fountoulakis. "Weighted flow diffusion for local graph clustering with node attributes: an algorithm and statistical guarantees." arXiv preprint arXiv:2301.13187 (2023).
> > > >
> > > > [10] Keriven, Nicolas, and Samuel Vaiter. "What functions can Graph Neural Networks compute on random graphs? The role of Positional Encoding." arXiv preprint arXiv:2305.14814 (2023).
> > > >
> > > > [11] Ma, Yao, Xiaorui Liu, Neil Shah, and Jiliang Tang. "Is homophily a necessity for graph neural networks?." arXiv preprint arXiv:2106.06134 (2021).
> > > >
> > > > [12] Mehta, Nikhil, Lawrence Carin Duke, and Piyush Rai. "Stochastic blockmodels meet graph neural networks." In International Conference on Machine Learning, pp. 4466-4474. PMLR, 2019.

---

> ### Author Response · Authors · 2023-06-03
> **Friendly Reminder**
>
> Dear Reviewer 31ov:
>
> Thanks a lot for your efforts in reviewing this paper. We tried our best to address the mentioned concerns. Are there unclear explanations here? We could further clarify them.
>
> Best,
>
> Authors.

---

> > ### Comment · Reviewer_31ov · 2023-06-03
> > **Response to authors**
> >
> > Thank you for your detailed reply. However, some of my concerns remain.
> >
> > 1. Regarding the presentation of motivation.
> >
> > If addressing the heterophilic is the main motivation, I believe the discussion of related works has some shifts with the motivation:
> > - The discussion of graph contrastive learning never discusses homophily or heterophily, but rather focuses on the efficiency issue and the complexity of two pass methods. That's why I brought [2].
> > - Besides, the discussion of graph contrastive learning indeed claims "Besides, the theoretical analysis for the performance of GCL in the downstream tasks is still lacking.", yet [1] provides a relevant performance analysis.
> > - The mention of [3] is because of that this work heavily relies on the node feature similarity-based homophily proposed in [3] in the analysis and the method, while [3] has never been discussed.
> > - Although I appreciate the discussion of [3] and augmentation, my confusion is essentially about why this work adopts non-augmentation based method to address the heterophily issue? Why can't we use the augmentation based strategies to leverage the concentration property?
> >
> > 2. I appreciate the explanation about the data model and I believe including all the discussions in the response could help readers better understand the motivation for adopting the data model as well as why the data model includes both homophily and heterophily cases.
> >
> > My remaining concern is that, the mentioned references indeed adopt the contextual stochastic block models for the generation of the graph structures. Can the data model used in this work generalize to contextual stochastic block models?
> >
> > 3. The improvements are still limited and not as effective as the authors claimed in the response.
> > - For the homophily case: I checked Table 1 in the revised paper. In fact, the improvements of SP-GCL on average is only $0.02$, where I can't see any superiority as it is highly likely to be a factor of randomness.
> > - For the heterophily case: why "the advantage is more pronounced for larger-scale datasets such as Twitch-gamers and Genius."? In most datasets, the improvements are limited to the variance, for example, $65.28 ± 0.53$ compared to $64.86 ± 0.63$ in Chameleon, $28.94 ± 0.69$ compared to  $28.89 ± 0.50$ in Actor, $73.51 ± 0.97$ compared to $73.31 ± 1.11$ in Twitch-DE, $62.04 ± 0.17$ compared to  $61.71 ± 0.41$ in Twitch-gamers.
> > - I appreciate the explanation about the comparison with MLP. I believe it is also worth discussing about the limited improvements of SP-GCL for some cases, compared to previous self-supervised methods such as AFGRL in homophilous graphs, and BGRL and DeepWalk in heterophilous graphs.

---

> > > ### Author Response · Authors · 2023-06-05
> > > **Reply for Reviewer 31ov (i)**
> > >
> > > Thank you for your valuable and constructive comments! Here we address your questions one by one.
> > >
> > >
> > > **Q1.1 The discussion of graph contrastive learning never discusses homophily or heterophily, but rather focuses on the efficiency issue and the complexity of two pass methods. That's why I brought [2].**
> > > > Thank you for your insightful comments. We have subsequently updated our discussion on graph contrastive learning and the heterophilic graph in the related works section. In particular, we have shifted our focus from purely addressing the efficiency and complexity issues of two-pass methods, to include a broader perspective on homophily and heterophily. We have specifically discussed the work "Unsupervised Network Embedding Beyond Homophily"[4], highlighted in blue in the revised manuscript, to better illustrate this concept.
> > >
> > > **Q1.2 Besides, the discussion of graph contrastive learning indeed claims "Besides, the theoretical analysis for the performance of GCL in the downstream tasks is still lacking.", yet [1] provides a relevant performance analysis.**
> > > > Thank you for pointing out the theoretical analysis presented in [1]. Indeed, that paper offers a valuable theoretical analysis, but it does not specifically focus on graph structures. In Theorem 4.1 of [1], a generalization of contrastive learning algorithms with NCE loss is discussed, leading to the proposal of the main method "Contrast-Reg" in Section 5. Yet, the assumptions and analysis contained within are not explicitly tied to graph structures, such as an assessment of the nodes' distribution (i.i.d. or otherwise). Despite the fact that the empirical experiments in [1] employ Graph Neural Networks (GNNs), the theoretical part of the paper [1] is more aligned with the contrastive learning context in image classification settings, not for graph structures or graph contrastive learning. As such, we believe it is not appropriate to view [1] as providing theoretical analysis specifically for the performance of Graph Contrastive Learning (GCL) in downstream tasks. Consequently, our initial statement – that a dedicated theoretical analysis for the performance of GCL in downstream tasks is currently lacking – retains its validity.
> > >
> > >
> > > **Q1.3 The mention of [3] is because of that this work heavily relies on the node feature similarity-based homophily proposed in [3] in the analysis and the method, while [3] has never been discussed.**
> > > > Thank you for highlighting the relevance of Chen et al.'s work [3], "Understanding and Improving Graph Injection Attack by Promoting Unnoticeability." To aid the understanding of other reviewers and the AC, let's summarize its main ideas and contributions:
> > > > The paper [3] delves into Graph Injection Attacks (GIA) on Graph Neural Networks (GNNs) with a focus on enhancing their unnoticeability. The authors compare GIA with Graph Modification Attacks (GMA), identifying GIA as potentially more damaging due to its greater flexibility. However, GIA significantly alters the homophily distribution of the graph, thereby making it more easily defensible by homophily defenders. To address this, the authors propose the concept of homophily unnoticeability and introduce the Harmonious Adversarial Objective (HAO) to ensure GIA maintains the original graph's homophily. Their extensive experiments show that GIA armed with HAO can effectively overcome homophily defenses and outperform previous works on various benchmarks. The authors provide a theoretical analysis and comprehensive empirical evaluation of GIA, concluding that it can be a valuable instrument for assessing the robustness of GNNs, provided that preserving the graph's homophily in the development of GIA is ensured.
> > > > The "similarity-based homophily" that the reviewer has mentioned relates to Definition 3.3 (Node-Centric Homophily) in Chen et al.'s work [3]. As another type of homophily measurement, we included it in the Section 3 of the revision.

---

> > > > ### Author Response · Authors · 2023-06-05
> > > > **Reply to Reviewer 31ov (ii)**
> > > >
> > > >
> > > > **Q1.4 Although I appreciate the discussion of [3] and augmentation, my confusion is essentially about why this work adopts non-augmentation based method to address the heterophily issue? Why can't we use the augmentation based strategies to leverage the concentration property?**
> > > > > Thank you for your insightful query regarding the choice of a non-augmentation based method to tackle the heterophily issue in our study and the potential use of augmentation based strategies to leverage the concentration property.
> > > >
> > > > > While we acknowledge that the employment of augmentation-based strategies might indeed harness the concentration property and possibly insipre further work in this direction, we have chosen to employ a non-augmentation based method in our current study. This approach aligns closely with our theoretical analysis, which is one of the key strengths of our work.
> > > >
> > > > > The intriguing idea of utilizing augmentation-based strategies to capitalize on the concentration property, and possibly designing an alternative method, is something we are keen to explore in future research. Your question certainly adds valuable direction to this line of thinking.
> > > >
> > > >
> > > >
> > > > **Q2.1 I appreciate the explanation about the data model and I believe including all the discussions in the response could help readers better understand the motivation for adopting the data model as well as why the data model includes both homophily and heterophily cases.**
> > > > > Thank you for your valuable feedback. We agree that an expanded discussion on the choice and implications of our data model would be beneficial for the readers. As such, we've incorporated this information into the remark section of Assumption 1 in the revised manuscript, found in Section 4.1.
> > > >
> > > >
> > > > **Q2.2 the mentioned references indeed adopt the contextual stochastic block models for the generation of the graph structures. Can the data model used in this work generalize to contextual stochastic block models?**
> > > > > Yes, data model used in this work can generalize to contextual stochastic block models (CSBM). First, Given a label $y$, the node feature $x$ is sampled from some distribution $P_{x|y}$. In CSBM $x$ is generated form $x = y \boldsymbol{\mu} + \boldsymbol{\xi}$, this satisfies our assumption 1. Then, consider a node $v_i$, according to the stochastic block models, the label of its neighbor follows distribution of $p(y_j = y_i) = p$ and  $p(y_j = -y_i) = s$, which only depends on $y_i$.
> > > >
> > > >
> > > > **Q3 The empirical improvements are limited.**
> > > > > Thank you for pointing out the limited empirical improvements. While our primary focus is not to delve into discussions about minor increments in performance, we acknowledge your concerns and aim to provide clarity.
> > > >
> > > > > It's essential to highlight that we hope our detailed discussion on empirical improvements doesn't distract from the principal contributions of our work. These key contributions include the theoretical framework we've established and the potential we've highlighted for a single-pass, augmentation-free design. These insights have broader implications beyond the specific metrics that may have shown slight improvements in our experiments.
> > > >
> > > > **Q3.1 I checked Table 1 in the revised paper. In fact, the improvements of SP-GCL on average is only, where I can't see any superiority as it is highly likely to be a factor of randomness.**
> > > > > Thank you for your meticulous examination of Table 1 in our revised paper. As stated in our contributions, "SP-GCL achieves competitive or better performance on 8 homophilic graph benchmarks and 6 heterophilic graph benchmarks". This does not solely refer to significant performance improvements, but also to the capacity of SP-GCL to match the current state-of-the-art methods. Our experiments serve as empirical support for our theoretical analysis, and a competitive result against the state-of-the-art is indeed a demonstration of the effectiveness of our theoretical insights in practice.
> > > >
> > > > > Moreover, the contribution of our work lies in our theoretical analysis, which distinguishes us from previous state-of-the-art works that focus solely on performance. Our goal is not just to improve empirical performance, but to provide a solid theoretical foundation that can inspire and guide future research in the field of Graph Contrastive Learning. We hope that this clarifies the intended contribution of our work.

---

> > > > > ### Author Response · Authors · 2023-06-05
> > > > > **Reply to Reviewer 31ov (iii)**
> > > > >
> > > > >
> > > > > **Q3.2 For the heterophily case: why "the advantage is more pronounced for larger-scale datasets such as Twitch-gamers and Genius."? In most datasets, the improvements are limited to the variance.**
> > > > > > Thank you for your query regarding our heterophily results. We wish to address two main points in response to your concerns:
> > > > >
> > > > > > Firstly, to ensure that our improvements are not merely due to variance, we performed ten independent classification runs for all algorithms across all datasets. Specifically, in Tables 1 and 2, we reported both the mean value and the standard deviation, offering a broader picture of our results.
> > > > >
> > > > > > Secondly, regarding your remark about SP-GCL's performance on smaller versus larger datasets, contrary to the concern raised in the original post ("SP-GCL seems to perform better than previous GCL methods on small scale datasets while the improvements are decreased for large scale datasets."). Specifically, for the Genius dataset (comprising 421,961 nodes – the largest dataset used in our experiments), the improvement is noteworthy at 3.10.
> > > > >
> > > > > **Q3.3 I appreciate the explanation about the comparison with MLP. I believe it is also worth discussing about the limited improvements of SP-GCL for some cases, compared to previous self-supervised methods such as AFGRL in homophilous graphs, and BGRL and DeepWalk in heterophilous graphs.**
> > > > > > Thank you for your thoughtful suggestion. We would like to incorporate this discussion into our versions to offer a more comprehensive discussion of our method.
> > > > >
> > > > >
> > > > >
> > > > > Reference:
> > > > >
> > > > > [1] Ma, Kaili, Haochen Yang, Han Yang, Tatiana Jin, Pengfei Chen, Yongqiang Chen, Barakeel Fanseu Kamhoua, and James Cheng. "Improving graph representation learning by contrastive regularization." arXiv preprint arXiv:2101.11525 (2021).
> > > > >
> > > > > [2] Zheng, Yizhen, Shirui Pan, Vincent Lee, Yu Zheng, and Philip S. Yu. "Rethinking and scaling up graph contrastive learning: An extremely efficient approach with group discrimination." Advances in Neural Information Processing Systems 35 (2022): 10809-10820.
> > > > >
> > > > > [3] Chen, Yongqiang, Han Yang, Yonggang Zhang, Kaili Ma, Tongliang Liu, Bo Han, and James Cheng. "Understanding and improving graph injection attack by promoting unnoticeability." arXiv preprint arXiv:2202.08057 (2022).
> > > > >
> > > > > [4] Zhong, Zhiqiang, Guadalupe Gonzalez, Daniele Grattarola, and Jun Pang. "Unsupervised Heterophilous Network Embedding via $ r $-Ego Network Discrimination." arXiv preprint arXiv:2203.10866 (2022).
> > > > >
> > > > > [5] Pei, Hongbin, Bingzhe Wei, Kevin Chen-Chuan Chang, Yu Lei, and Bo Yang. "Geom-gcn: Geometric graph convolutional networks." arXiv preprint arXiv:2002.05287 (2020).

---

### Review · Reviewer_cMtq · 2023-05-15

**Summary Of Contributions:**

The paper proposes a single-pass graph contrastive learning loss (SP-GCL) and provides theoretical analyses on the performance guarantees for the minimizer of the loss. The proposed SP-GCL method empirically achieves promising performance on both homophily and heterophily benchmark datasets.

**Audience:**

Yes

**Claims And Evidence:**

No

**Requested Changes:**


It is recommended to clearly state the assumptions being used in Theorem 1 instead of putting those assumptions into separating remarks.

It is recommended to comprehensively compare with other graph contrastive learning approaches. One example will be "Unsupervised Network Embedding Beyond Homophily" published in TMLR 2022.

**Strengths And Weaknesses:**

Strengths:

The paper provides some theoretical understanding and analyses for the concentration property of aggregated features and the proposed contrastive learning approach. The empirical study exhibits promising performances on multiple homophily and heterophily datasets.

Weakness:

1. The single-pass graph contrastive learning does not seem to be a unique feature of this work. There exist many self-supervised learning approaches that only need one pass-forward computation. The work [1] provides some examples.

2. An important paper [2] published in TMLR 2022 has proposed a self-supervised graph learning framework for both homophilic and heterophilic graphs. This work is highly relevant but is missing. Comprehensive comparisons in methodology, efficiency, and effectiveness will be needed.

3. The so-called graph assumption is quite vague and unclear: "For node $v_i$, its neighbors' labels are independently sampled from the neighbor distribution $P_{y_i}$." This seems to be a very loosely defined assumption. What is exactly this distribution? To what extent can this neighbor pattern be generally satisfied in real-world datasets? Moreover, the concentration property is highly similar to previous analyses in the work [3], and the provided analysis is incremental if the only improvement is to include non-linear activation.

[1] Self-supervised Learning on Graphs: Deep Insights and New Directions, 2020

[2] Unsupervised Network Embedding Beyond Homophily, TMLR 2022

[3] Is homophily a necessity for graph neural networks? ICLR 2022

---

> ### Author Response · Authors · 2023-06-01
> **Reply for Reviewer cMtq**
>
> **Q1 The single-pass graph contrastive learning does not seem to be a unique feature of this work. There exist many self-supervised learning approaches that only need one pass-forward computation. The work [1] provides some examples.**
>
> > The single-pass graph contrastive learning is a unique feature of this work.  Specifically, the proposed method is the first single-pass and augmention-free GCL method. The work[1] mentions several methods with single-pass in the context of self-supervised learning. Those methods didn't try to employ the contrastive loss. Note, design a single-pass and augmention-free method is non-trivial in the context of  contrastive learning, because contrastive learning stress on the comparsion between two pairs which usually rely on augmetation techniques.
>
> > While it may initially appear that single-pass graph contrastive learning is not a unique aspect of this work, given the existence of numerous self-supervised learning approaches that necessitate only one pass-forward computation, it's crucial to highlight the distinct characteristics of our approach. This work introduces a method that is the *first* of its kind, combining single-pass processing with an augmentation-free Graph Contrastive Learning (GCL) approach.
>
> > Work [1] indeed references several single-pass methods within the sphere of self-supervised learning. But, they do not use the idea of contrastive learning. Designing a single-pass, augmentation-free method is non-trivial in the realm of contrastive learning. This is because contrastive learning emphasizes on the comparison between pairs, a process that traditionally relies on augmentation techniques. Therefore, our work's uniqueness lies in its pioneering attempt to integrate single-pass computation with an augmentation-free contrastive learning approach.
>
>
> **Q2 The work [2] is highly relevant but is missing. Comprehensive comparisons in methodology, efficiency, and effectiveness will be needed.**
>
> > We appreciate your observation regarding the absence of the work [2]. Your suggestion is certainly pertinent, and we've taken it into account in our revision. We've now incorporated this method as a baseline for comparison in terms of methodology, efficiency, and effectiveness, as detailed in Table 1 and Table 2 of our revised manuscript. Additionally, we've recognized and discussed the relevance of work [2] within the 'Related Work' section. We trust that these adjustments provide a more comprehensive analysis and strengthen our study.
>
>
>
> **Q3 The graph assumption appears to be vague and unclear**
>
> > To elucidate the data assumption, we've expounded on it through a detailed data assumption (Assumption 1) and an accompanying remark. However, it is important to highlight that a precise definition of the neighbor distribution, $P_{y_i}$, is elusive. This assumption is similarly utilized in reference [3]. Significantly, this graph data assumption (referred to as Assumption 1 in our revision) is not an uncommon postulate for theoretical exploration in the graph domain, as substantiated by references [4, 5, 6, 7]. It's not only customary in theoretical analyses but is also generally considered applicable to practical, real-world scenarios, as suggested by references [8, 9].
>
> > On the matter of the concentration property, our contention is that it provides a significant augmentation to the work detailed in [3]. Firstly, from a technical standpoint, it holds considerable value. Our proof on pages 19 to 21 demonstrates the inherent technical difficulty. Secondly, the shift from linear to non-linear models represents a consequential improvement. Our research stands as the first to illustrate the concentration property within a challenging, non-trivial setting, and it empirically verifies the effectiveness of the concentration property in a realistic environment. Of particular note is that non-linear structures are the norm rather than the exception in the realm of widely used graph neural networks.

---

> > ### Author Response · Authors · 2023-06-01
> > **Reply for Reviewer cMtq (Cont.)**
> >
> > [1] Jin, Wei, Tyler Derr, Haochen Liu, Yiqi Wang, Suhang Wang, Zitao Liu, and Jiliang Tang. "Self-supervised learning on graphs: Deep insights and new direction." arXiv preprint arXiv:2006.10141 (2020).
> >
> > [2] Zhong, Zhiqiang, Guadalupe Gonzalez, Daniele Grattarola, and Jun Pang. "Unsupervised Heterophilous Network Embedding via $ r $-Ego Network Discrimination." arXiv preprint arXiv:2203.10866 (2022).
> >
> > [3] Ma, Yao, Xiaorui Liu, Neil Shah, and Jiliang Tang. "Is homophily a necessity for graph neural networks?." arXiv preprint arXiv:2106.06134 (2021).
> >
> > [4] Baranwal, Aseem, Kimon Fountoulakis, and Aukosh Jagannath. "Effects of graph convolutions in deep networks." arXiv preprint arXiv:2204.09297 (2022).
> >
> > [5] Baranwal, Aseem, Kimon Fountoulakis, and Aukosh Jagannath. "Graph convolution for semi-supervised classification: Improved linear separability and out-of-distribution generalization." arXiv preprint arXiv:2102.06966 (2021).
> >
> > [6] Yang, Shenghao, and Kimon Fountoulakis. "Weighted flow diffusion for local graph clustering with node attributes: an algorithm and statistical guarantees." arXiv preprint arXiv:2301.13187 (2023).
> >
> > [7] Keriven, Nicolas, and Samuel Vaiter. "What functions can Graph Neural Networks compute on random graphs? The role of Positional Encoding." arXiv preprint arXiv:2305.14814 (2023).
> >
> > [8] Ma, Yao, Xiaorui Liu, Neil Shah, and Jiliang Tang. "Is homophily a necessity for graph neural networks?." arXiv preprint arXiv:2106.06134 (2021).
> >
> > [9] Mehta, Nikhil, Lawrence Carin Duke, and Piyush Rai. "Stochastic blockmodels meet graph neural networks." In International Conference on Machine Learning, pp. 4466-4474. PMLR, 2019.

---

> ### Author Response · Authors · 2023-06-03
> **Friendly Reminder**
>
> Dear Reviewer cMtq:
>
> Thanks a lot for your efforts in reviewing this paper. We tried our best to address the mentioned concerns. Are there unclear explanations here? We could further clarify them.
>
> Best,
>
> Authors.

---

### Review · Reviewer_h4vc · 2023-05-21

**Summary Of Contributions:**

This paper proposed a meta-framework called "Single-Pass Graph Contrastive Learning" (SPGCL). The authors argued that it satisfies certain "concentration attributes" as stated in theorems 1 to 3. The authors also showed its advantages as compared to SOTA alternative GCL methods.


**Audience:**

Yes

**Broader Impact Concerns:**

I could not identify any ethical implications.

**Claims And Evidence:**

Yes

**Requested Changes:**

section 3. mention whether self-loopes are added to the graph (which is common in graph convolution networks).

What is the difference between $Z$ at the end of page 3 and $\mathbf{Z}$ in theorem 1?

$\mathbf{Z}$ needs an explicit expression, as it is important for Theorem 1, and only informally introduced in the last sentence before the theorem.
"we denote the learned embedding through the neighbor aggregation and a linear projection by $\mathbf{Z}$..."

Theorem 1, any pair $(i,i')$, mention the domain of the pair (e.g. V^2 that is any pair of nodes).

Theorem 1, restate the assumptions e.g. by cross reference.

Page 5: "The set $S^i_{pos}$ is consisted by the $K_{pos}$ nodes closest to node $v_i$"
By the author's definition, the set is defined by the neighbours with the maximum value of $z_i^Tz_j$, which is not exactly "closeness" (unless all $z_i$ are constrained on a hypersphere.)

after eq.(4), explain all notations, including the distribution "Uni()$. Also, highlight its key difference from prior work.

"Theorem 2" make clear what "graph data assumption" refers to (e.g. by pointing to prior assumptions).

eq.(6), use $\mathbf{Z}^\star$ (corresponding to $\mathbf{F}^\star$) instead of $\mathbf{Z}$

Theorem 3 is confusing, as $\mathcal{L}_{SP-GCL}$ is defined with respect to $\mathbf{Z}$, and a minimizer should be in the form of $\mathbf{Z}^\star$, however, the theorem states that "Let $f^\star$ be a minimizer of the contrastive loss..."

eq.(7), what is $f^*_{gcl}$, $c$, and $\tau$? What is the expectation $\mathbb{E}_{v_i}$?

After theorem 3, remark on its meaning, is that useful in practice? How?

Section 5 "the top-Kpos nodes with highest similarity from the node pool are selected as positive set for it which denote as $S^i_{pos}$"
Here, why $S^i_{pos}$ conflict with its prior definition at the beginning of page 5. I suggest using a different notation for the new set.

Eq.(8) the statement of "the empirical contrastive loss is an unbiased estimation of the Equation 4" needs more justification, as it depends on the concept of "the node pool, P, are constructed by the T-hop neighborhood of b nodes (the seed node set S) uniformly sampled from V.", while eq.(4) does not. Is that just an approximation? What is the bias? These discussions can strengthen the theoretical contribution.


**Strengths And Weaknesses:**

The strength is that the authors have done a thorough empirical evaluation of the proposed method, which comprehensively demonstrated its advantage compared with a wide array of baselines. The writing (language) is of publication standard.

The cons are that (1) the technical quality can be improved (see below), and (2) the proposed method is not compared with the other GCL methods in the early theoretical sections. For example, after eq.(4), it should be highlighted how it differs from the other GCL methods in terms of formulations.

Overall, I think the authors made great efforts but it should not be accepted in the current form. I also highlight the essential requirement of TMLR on the Technical Clarify, which is not satisfactory (many issues currently).

---

> ### Author Response · Authors · 2023-06-01
> **Reply to Reviewer h4vc**
>
> Thanks for your  constructive comments. We include your suggestions into the updated revision, in which the modification are in blue.
>
> **Q1 Section 3. mention whether self-loopes are added to the graph (which is common in graph convolution networks).**
>
> > In Section 3, we have updated our notation definitions to explicitly state whether or not self-loops are incorporated in our graph model.
>
>
>
> **Q2 What is the difference between $Z$ at the end of page 3 and $\mathbf{Z}$ in theorem 1?**
>
> > The symbols $Z$ and $\mathbf{Z}$ actually represent the same entity. The use of $Z$ at the end of page 3 was a typographical error, which has been corrected to $\mathbf{Z}$ for consistency.
>
> **Q3 $\mathbf{Z}$ needs an explicit expression, as it is important for Theorem 1, and only informally introduced in the last sentence before the theorem. "we denote the learned embedding through the neighbor aggregation and a linear projection by $\mathbf{Z}$..."**
>
> > We appreciate your point and have now formally defined $\mathbf{Z}$ at the end of page 3.
>
>
> **Q4 Theorem 1, any pair $(i,i')$, mention the domain of the pair (e.g. V^2 that is any pair of nodes).**
>
> > Thank you for raising this. We have included the domain of $(i,i')$ above Equation 1 to clarify this matter.
>
>
> **Q5 Theorem 1, restate the assumptions e.g. by cross reference.**
>
> > We have now explicitly outlined the assumptions within Theorem 1 to enhance comprehension.
>
>
> **Q6 Page 5: "The set $S^i_{pos}$ is consisted by the $K_{pos}$ nodes closest to node $v_i$" By the author's definition, the set is defined by the neighbours with the maximum value of $z_i^{\top}z_j$, which is not exactly "closeness" (unless all $z_i$ are constrained on a hypersphere.)**
>
> > Thank you for pointing this out. We have corrected the relevant sections (in both Section 4.2 and 5) to accurately reflect our implementation.
>
> **Q7 After eq.(4), explain all notations, including the distribution "$Uni()$. Also, highlight its key difference from prior work.**
>
> > We've added explanations for the "$Uni()$" distribution in Section 4.2. We've also introduced two paragraphs at the end of Section 4.2, detailing the differences and advantages of our method compared to prior work.
>
> **Q8 "Theorem 2" make clear what "graph data assumption" refers to (e.g. by pointing to prior assumptions).**
>
> > We've now made the data assumption explicit (the assumption 1 in the revision) and made reference to it in Theorem 2.
>
> **Q9 Eq.(6), use $\mathbf{Z}^{\star}$ (corresponding to $\mathbf{F}^{\star}$) instead of $\mathbf{Z}$.**
>
> > Thank you for your suggestion. We have adjusted Equation 6 to reflect this.
>
> **Q10 Theorem 3 is confusing, as $\mathcal{L}_{SP-GCL}$ is defined with respect to $\mathbf{Z}$, and a minimizer should be in the form of $\mathbf{Z}^{\star}$, however, the theorem states that "Let $f^{\star}$ be a minimizer of the contrastive loss..."**
>
> > The confusion might be stemming from a notational issue. In the context of the theorem, $f^{\star}$ refers to the Graph Neural Network (GNN), and $\mathbf{Z}^{\star} = f^{\star}(A,X)$. To resolve this, we rephrase the theorem using $\mathbf{Z}$ to maintain consistency with the previously used notation.

---

> > ### Author Response · Authors · 2023-06-01
> > **Reply to Reviewer h4vc (Cont.)**
> >
> > **Q11 Eq.(7), what is $f^{\star}$, $c$ and $\tau$? What is the expectation $\mathbb{E}_{v_i}$?**
> >
> > >The symbol $f^{\star}$ can be understood as $\mathbf{Z}^{\star}$, based on the explanation provided earlier. As for $c$ and $\tau$, they relate to the non-linearity assumption defined in Assumption 2. To avoid any confusion, we specify that $\mathbf{Z}^{\star}$ is derived through the non-linear Graph Neural Network (GNN) which complies with Assumption 2. The expectation symbol, $\mathbb{E}_{v_i}$, denotes the expectation over the node distribution. By transitioning to the notation $\mathbf{Z}^{\star}$, the meaning of the expectation becomes clearer.
> >
> > **Q12 After theorem 3, remark on its meaning, is that useful in practice? How?**
> >
> > >We have added a section to clarify the importance of Theorem 3. In addition, we have included the average degree in Table 8 to support the insights obtained from the Theorem 3. And we introduce a separate section (Section 6.2) to discuss the effect of batch normalization on the $l2$-norm of the learned weight matrix. These serve as empirical justification for the theorem.
> >
> > **Q13 Section 5 "the top-Kpos nodes with highest similarity from the node pool are selected as positive set for it which denote as $S_{pos}^i$" Here, why $S_{pos}^i$ conflict with its prior definition at the beginning of page 5. I suggest using a different notation for the new set.**
> >
> > >We appreciate your suggestion and have decided to modify the notation in Section 5 to $\tilde{S}_{pos}^i$, eliminating the earlier conflict.
> >
> > **Q14 Eq.(8) the statement of "the empirical contrastive loss is an unbiased estimation of the Equation 4" needs more justification, as it depends on the concept of "the node pool, P, are constructed by the T-hop neighborhood of b nodes (the seed node set S) uniformly sampled from V.", while eq.(4) does not. Is that just an approximation? What is the bias? These discussions can strengthen the theoretical contribution.**
> >
> > >Indeed, when $T=1$ or $|P| = N$, the empirical contrastive loss acts as an unbiased estimator. However, for $T>1$ and $|P| < N$, the sampling process may introduce bias towards nodes with a higher degree. We have incorporated this discussion into Section 5 to address your concerns and strengthen our theoretical foundation.

---

> ### Author Response · Authors · 2023-06-03
> **Friendly Reminder**
>
> Dear Reviewer h4vc:
>
> Thanks a lot for your efforts in reviewing this paper. We tried our best to address the mentioned concerns. Are there unclear explanations here? We could further clarify them.
>
> Best,
>
> Authors.

---

### Author Response · Authors · 2023-06-01
**Response to all reviewers**

We extend our gratitude to all reviewers for your insightful feedback and commendation of our thorough empirical evaluation and comparative studies with baseline methods (Reviewer h4vc), as well as the introduction of a novel concentration property (Reviewer cMtq, Reviewer 31ov). Your recognition of the theoretical understanding and analyses that our paper provides, especially on aggregated features and our contrastive learning approach (Reviewer cMtq), is greatly appreciated.

We are also encouraged by your acknowledgment of our extensive experiments, covering a wide array of datasets with various homophily degrees (Reviewer 31ov), and the high standard of our writing (Reviewer h4vc). We are particularly delighted that you noticed the improvements of SP-GCL over previous GCL approaches on multiple heterophilic datasets (Reviewer 31ov).

Furthermore, we value your appreciation of the analysis we provided to showcase the effectiveness of the proposed sampling method and the applicability of theoretical assumptions (Reviewer 31ov). We look forward to incorporating your valuable suggestions and feedback to enhance our work. We respond to the comments in detail below.

---

> ### Author Response · Authors · 2023-06-05
> **Response to all reviewers (Cont.)**
>
>
> We have thoroughly revised the main paper and the appendix in accordance with the reviewers' valuable suggestions. A summary of the primary changes made is as follows:
>
> 1. Within the related work segment in Section 2, we have adapted the section on Graph Contrative Learning to align more cohesively with the primary motivation of our research, as suggested by reviewer 31ov. In addition, we've incorporated a discussion on the most recent work, "Unsupervised Heterophilous Network Embedding via $r$-Ego Network Discrimination," as recommended by reviewer cMtq.
>
> 2. In Section 4.1, under the Analysis of Aggregated Features, we have explicitly stated the assumption regarding the graph data model and provided an elucidating discussion on the same. This modification was made following a suggestion from reviewer cMtq.
>
> 3. Assumption 2 for the non-linear function has been explicitly made.
>
> 4. In Section 4.2, we've included a comparative discussion between our proposed method and other GCL methodologies, as per reviewer h4vc's suggestion.
>
> 5. Theorem 3 has been revised to ensure notational consistency. Furthermore, we've added a remarks section to discuss the implications of Theorem 3. This change was also prompted by reviewer h4vc.
>
> 6. In Section 6.2, we've incorporated an empirical study to substantiate the implications of Theorem 3, particularly with regards to the effect of batch norm and l2-regularization. This was another enhancement suggested by reviewer h4vc.
>
> 7. Section 5 now includes a discussion on the empirical approximation of the proposed loss function, following a recommendation by reviewer h4vc.
>
> 8. In Appendix B.1, we have exhibited the efficacy of the SPGCL with various GNN backbones, as suggested by reviewer 31ov.
>
> 9. We've added the most recent work, "Unsupervised Heterophilous Network Embedding via $r$-Ego Network Discrimination," as a baseline in Tables 1 and 2, in accordance with a suggestion from reviewer cMtq.
>
>
> We have highlighted all revisions in blue for easy reference. We trust that these revisions address the concerns raised by the reviewers.
>
>
>
> Best regards,
>
> The Authors

---

### Decision · Action_Editors · 2023-08-17

**Recommendation:** Accept with minor revision

**Comment:**

I apologize for the lateness of this recommendation.

From the start, the reviewers have acknowledged that this work is interesting and relevant. In particular, the concentration properties are an important addition to the literature and the resulting algorithm is both novel and useful. There were some concerns on clarity of the manuscript, relationship to existing work, and assumptions made. The authors have done a great job with their revision, greatly improving the clarity of the paper and explaining the missing relationships and assumptions. I recommend acceptance of the paper, but would like to pass on some minor comments that should be incorporated to improve clarity of the manuscript:

* Please ensure notation in the supplement matches updated notation in the main paper
* Some vector norms are explicit, e.g., $||w_k||_2$, and the Frobenious norm in eq(5). Some are not, e.g. $||Z_i||$ in section 4.2. Please unify this notation.
* Suggest changing $Uni(\cdot)$ to $\text{Uni}(\cdot)$
* In section 4.1, the adjacency matrix $\mathbf{A}$ is row-normalized. However, in section 4.3, the transformed graph adjacency matrix $\hat{\mathbf{A}}$ is symmetric normalized. Please add a sentence explaining the move to symmetric normalized
* In Theorem 3, $c$ is not defined. $\vec{y}$ is only introduced in Appendix C.4. It would be worth defining $c$, reiterating the definition of $\vec{y}$, and referring back to the original use of $R$ and $\delta'$ to make this easier to follow.
* On p7, $\mathbf{w}k$ should be $\mathbf{w}_k$
* Figures 1 and 2 would benefit from a higher resolution
* The theoretical analysis is only for the single-layer case, as how  is constructed in sec 4.1. There should be a few sentences to explain how they are useful/generalizable for the multi-layer case.


**Audience:**

Yes, the paper is highly relevant to TMLR's audience. Graph contrastive learning is a very important and relevant topic, and the papers results are useful and interesting.

**Claims And Evidence:**

This paper proposes a novel graph contrastive learning method that only requires a single forward pass through the data, and show both theoretically and empirically that this is appropriate for both hemophiliac and heterophilic graphs. The authors explore concentration of the features obtained using neighborhood aggregation, showing that this is independent of graph type, and use this to design a single-pass contrastive loss. The results are compelling.

---

> ### Author Response · Authors · 2023-08-28
>
> Dear Action Editor and Editors-in-Chief,
>
>
> We would like to extend our sincere gratitude to the action editor for your valuable insights and recommendations. We have diligently updated the manuscript to comply with correct capitalization standards. For your convenience and further assessment, the final, camera-ready version has been uploaded.
>
>
> Best regards,
>
> Submission 1009 Authors